# Deep Forcing: Training-Free Long Video Generation with Deep Sink and Participative Compression

**Jung Yi** [1]  **Wooseok Jang** [1]  **Paul Hyunbin Cho** [1]  **Jisu Nam** [1]  **Heeji Yoon** [1]  **Seungryong Kim** [1]

**Project Page**: https://cvlab-kaist.github.io/DeepForcing/

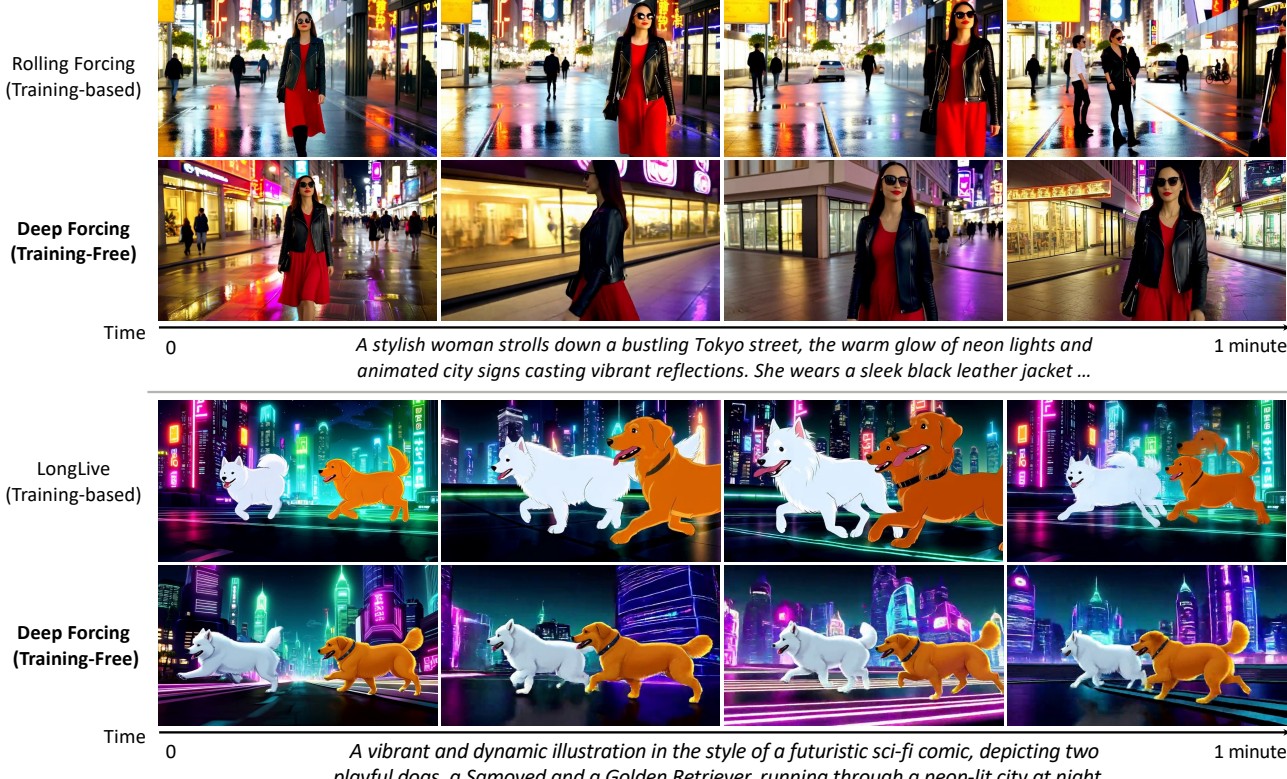

Rolling Forcing (Training-based)

**Deep Forcing (Training-Free)**

Time  0  *A stylish woman strolls down a bustling Tokyo street, the warm glow of neon lights and animated city signs casting vibrant reflections. She wears a sleek black leather jacket …*  1 minute

LongLive (Training-based)

**Deep Forcing (Training-Free)**

Time  0  *A vibrant and dynamic illustration in the style of a futuristic sci-fi comic, depicting two playful dogs, a Samoyed and a Golden Retriever, running through a neon-lit city at night…*  1 minute

*Figure 1.* Our ***training-free*** approach, **Deep Forcing**, achieves comparable visual quality with more dynamic motion to even ***training-based*** baselines, such as Rolling Forcing (Liu et al., 2025) and LongLive (Yang et al., 2025b).

## Abstract

Recent advances in autoregressive video diffusion have enabled real-time frame streaming, however, existing methods still suffer from visual error accumulation including visual fidelity and motion degradation over long-horizon. To address these challenges, we introduce **Deep Forcing**, a **training-free** extension of autoregressive video diffusion models that stabilizes long video generation through two complementary mechanisms.

[1]KAIST AI. Correspondence to: Seungryong Kim <seungryong.kim@kaist.ac.kr>.

*Proceedings of the $43^{rd}$ International Conference on Machine Learning*, Seoul, South Korea. PMLR 306, 2026. Copyright 2026 by the author(s).

**Deep Sink** preserves approximately half of the sliding context window as persistent sink tokens and realigns their temporal RoPE phases to the current timeline, thereby maintaining global context during extended rollouts. **Participative Compression** performs importance-aware KV cache pruning, retaining only tokens that actively participate in recent attention while removing redundant or degraded history, effectively mitigating error accumulation under out-of-distribution lengths. Together, these components enable **over** $12\times$ **length extrapolation** (e.g., **5s-trained → 60s+**) without sacrificing inference speed, while improving visual fidelity and motion dynamics compared to prior methods. Our results demonstrate that **Deep Forcing** can achieve performance compa-

rable to state-of-the-art training-based methods trained specifically for long video generation.

## 1. Introduction

Recent advances in video diffusion models (Yang et al., 2024; Wan et al., 2025) have demonstrated remarkable capability in synthesizing short video clips (e.g., 81 frames) with high visual fidelity and coherent motion dynamics.

Building on this progress, emerging interactive world models (Parker-Holder & Fruchter, 2025; rre, 2025; Ye et al., 2025; Shin et al., 2025) require autoregressive long-horizon video generation (e.g., 1–2 minutes), where frames are generated and streamed sequentially in real time. Unlike conventional offline video generation, which synthesizes an entire clip at once, autoregressive video generation operates in an online manner: each frame is generated, emitted, and immediately consumed before the next frame is produced.

Self-Forcing (Huang et al., 2025) and its variants (Yang et al., 2025b; Liu et al., 2025; Cui et al., 2025; Teng et al., 2025; Yin et al., 2025) have become standard in autoregressive video generation. These approaches typically employ causal attention masks and maintain historical frames in the key–value (KV) cache, while using self-generated historical frames during training to reduce the train–test distribution gap.

Nevertheless, such methods still remain vulnerable to error accumulation over long horizons, leading to progressive degradation in visual fidelity, where colors drift toward oversaturation, textures become blurred, and fine-grained details gradually vanish. We attribute this failure primarily to the First-In–First-Out (FIFO) KV-cache strategy. Specifically, FIFO (1) overly relies on the most recently generated tokens, which tend to contain the largest accumulated errors, and (2) indiscriminately evicts the earliest tokens regardless of their relevance to the current generation, often discarding the most informative context.

To address this, in this paper, we propose **Deep Forcing**, a novel **tuning-free** method that further mitigates error accumulation in long-horizon video generation on top of pretrained Self Forcing. Our approach enables minute-long video generation that preserves both visual fidelity and motion stability, without requiring any fine-tuning.

Specifically, we observe that pre-trained Self Forcing (Huang et al., 2025) attends not only to recently generated tokens, but also to the earliest and intermediate tokens across the sequence. Motivated by this observation and recent advances in large language models (LLMs) (Ghadia et al., 2025; Shi et al., 2025; Li et al., 2024), we introduce **Deep Sink**, which (i) preserves a large fraction of the earliest tokens in the KV cache (40–60%) to maintain globally

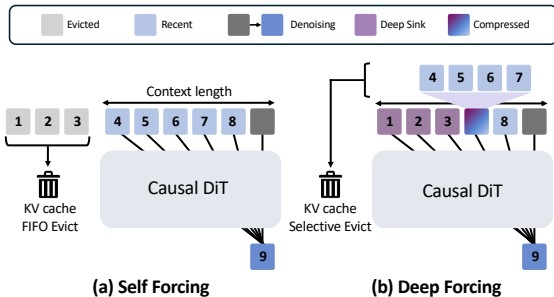

*Figure 2.* **Comparison of KV-Cache Management.** (a) **Self Forcing** (Huang et al., 2025) adopts a FIFO policy that discards the earliest tokens regardless of their importance, often removing critical contextual information and degrading visual fidelity over long-horizon. In contrast, (b) **Deep Forcing** preserves **Deep Sink** tokens while applying KV-cache compression via **Participative Compression**, effectively improving visual quality and motion stability over long-horizon generation.

informative context, and (ii) dynamically adjusts the Relative Positional Embedding (RoPE) to support long-horizon video generation.

While Deep Sink improves visual fidelity over long-horizon, it still suffers from error accumulation during minute-long video generation (e.g., up to $12\times$ longer than the training clip length). To address this, we further present **Participative Compression**, an importance-aware KV-cache compression strategy that removes tokens irrelevant to the currently generated frames, thereby reducing redundancy and mitigating fidelity degradation caused by noise accumulation from outdated tokens.

Despite its simplicity, Deep Forcing outperforms even training-based methods (Liu et al., 2025; Yang et al., 2025b), as shown in Fig. 1. Comprehensive evaluations using VBench (Huang et al., 2024), user studies, and Vision Language Model (VLM) assessments show that our training-free approach significantly improves both visual fidelity and motion dynamics, consistently outperforming strong training-based baselines (Liu et al., 2025; Yang et al., 2025b). Ablation studies further validate the effectiveness of our design choices.

Our contributions are summarized as follows:

- We propose **Deep Forcing**, a **tuning-free** framework that significantly mitigates error accumulation in long-horizon autoregressive video generation on top of pretrained Self Forcing and its variants.

- We introduce **Deep Sink**, which preserves a large portion of the earliest tokens in the KV cache while dynamically adjusting relative positional embeddings.

- We present **Participative Compression**, the first training-free KV-cache compression method for autoregressive video diffusion, which removes redundant tokens while preserving the most relevant context.

- Extensive evaluations on VBench, user studies, and

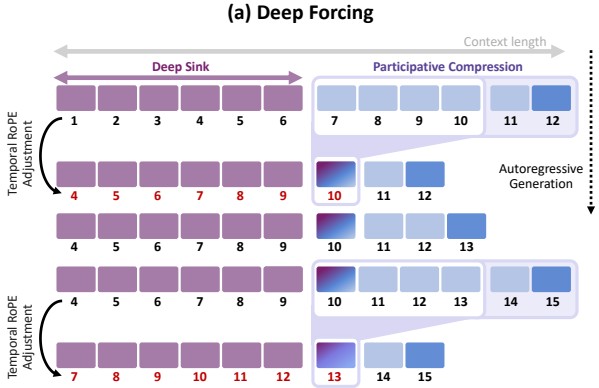

*Figure 3.* **Overview of Deep Forcing.** (a) **Deep Sink** maintains a substantially enlarged attention sink that covers approximately half of the context window, while applying Participative Compression to the remaining rolling portion. A temporal RoPE adjustment aligns the temporal indices of sink tokens with the current frames, thereby preserving temporal coherence. (b) **Participative Compression** computes query-averaged attention scores between recent tokens and candidate tokens, selecting the top-$C$ most important tokens to retain in the compressed KV cache while evicting redundant tokens.

VLM assessments demonstrate that our training-free approach achieves performance comparable to state-of-the-art training-based methods.

## 2. Related Work

**Autoregressive Video Diffusion.** A growing line of work (Chen et al., 2024; Teng et al., 2025; Huang et al., 2025; Yin et al., 2025) combines diffusion modeling with autoregressive (AR) prediction to support long-horizon or streaming video generation. CausVid (Yin et al., 2025) converts a pre-trained bidirectional diffusion transformer into a causal AR generator with KV cache. Building on these ideas, Self Forcing (Huang et al., 2025) addresses the train–inference mismatch by conditioning the model on its own generated frames. Rolling Forcing (Liu et al., 2025) proposes expanding the diffusion window, and LongLive (Yang et al., 2025b) incorporates KV recaching to maintain visual continuity while ensuring prompt adherence across scene transitions. In contrast, our method is tuning-free, leveraging the inherent attention-sink behavior of pre-trained Self Forcing to achieve comparable performance to training-based methods.

**Attention Sink.** Recent work has revealed that attention in autoregressive models concentrates disproportionately on initial tokens, termed attention sinks (Xiao et al., 2023). StreamingLLM (Xiao et al., 2023) showed that retaining these sink tokens within a sliding window enables robust generation beyond the training context length. Recent autoregressive video models (Yang et al., 2025b; Liu et al., 2025) maintain the first three frames as attention sinks via model distillation or fine-tuning. We demonstrate that the pre-trained autoregressive video diffusion model (Huang et al., 2025) exhibits inherent attention sink behavior that can be effectively leveraged without training, requiring deeper context preservation.

**KV Cache Compression.** The linearly growing KV cache in autoregressive generation motivates compression strategies that reduce memory footprint while preserving generation quality. As the cache grows, attention becomes distributed across increasingly many tokens, diluting focus on critical context and degrading output quality. To address this, recent works employ attention-based token selection for long-context LLM generation. H2O (Zhang et al., 2023) and SnapKV (Li et al., 2024) preserve important tokens based on cumulative attention scores and observation windows. MorphKV (Ghadia et al., 2025) maintains constant-size caches through correlation-aware ranking. While these methods target language models, similar memory constraints arise in autoregressive video diffusion, where temporal context must be efficiently maintained across frames. We extend these principles through Participative Compression.

## 3. Preliminaries

**Autoregressive Video Diffusion.** Autoregressive video diffusion models (Chen et al., 2025; Huang et al., 2025) produce each frame or chunk conditioned on previously generated frames or chunks within a denoising diffusion process.

Specifically, given a video sequence of $N$ frames $x^{1:N} = (x^1, x^2, \ldots, x^N)$, the autoregressive model applies the chain rule to factorize the joint distribution as

$$p(x^{1:N}) = \prod_{i=1}^{N} p(x^i \mid x^{<i}). \tag{1}$$

A diffusion model parameterizes each conditional $p(x^i \mid x^{<i})$, generating the $i$-th frame by conditioning on previously generated frames $x^{<i} = (x^1, x^2, \ldots, x^{i-1})$.

**Self Forcing.** Self Forcing (SF) (Huang et al., 2025) generates videos in an autoregressive manner using a First-In–First-Out (FIFO) Key-Value (KV) cache mechanism,

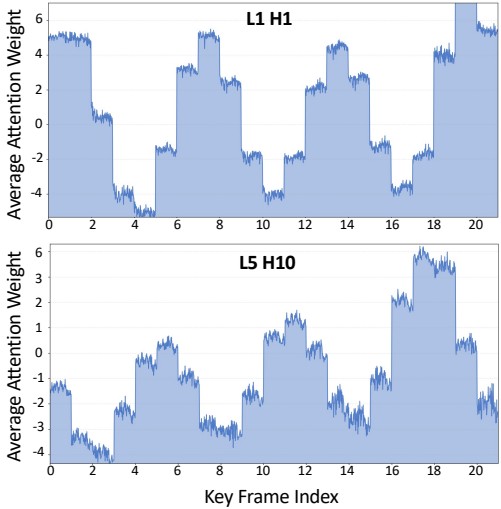

Figure 4. **Attention weight distribution.** Query-averaged attention score showing how the last frames (frames 19-21) attends to earlier frames in KV cache (frames 0-18). We visualize two representative attention heads from different layers, L1H1 (layer 1, head 1) and L5H10 (layer 5, head 10), demonstrating that substantial attention is maintained across the entire context window, not just initial frames. We provide additional layer/head analyses in Appendix T.

producing frames sequentially while evicting the earliest generated tokens from the KV cache.

Specifically, SF maintains a fixed-size KV cache of length $L$, which stores KV pairs corresponding to the most recent $L$ generated frames. When the cache reaches its capacity, the oldest entries are evicted to accommodate newly generated frames. During generation, self-attention is computed between the queries of the frame(s) currently being generated and the keys and values stored in the KV cache from previously generated frames.

SF employs a multi-step denoising process with the noise schedule $\{t_0 = 0, t_1 = 250, t_2 = 500, t_3 = 750, t_4 = 1{,}000\}$ ($T = 4$), corresponding to five noise levels. Each frame $i$ is denoised iteratively across these timesteps. At denoising step $t_j$, the diffusion model $G_\theta$ predicts a denoised latent $x_{t_{j-1}}^i$ from the noisy input $x_{t_j}^i$, conditioned on the KV cache of previously generated clean frames and the current timestep $t_j$. Formally, this process is given by:

$$x_{t_{j-1}}^i = \Psi\big(G_\theta(x_{t_j}^i, t_j, KV), t_{j-1}\big), \qquad (2)$$

where $x_{t_4}^i \sim \mathcal{N}(0, I)$ and $KV$ denotes the KV cache constructed from previously generated frames.

## 4. Method

We propose a training-free method, **Deep Forcing**, that reduces error accumulation in long video generation based on pre-trained Self Forcing (Huang et al., 2025). Our approach consists of two core components: **Deep Sink**, which

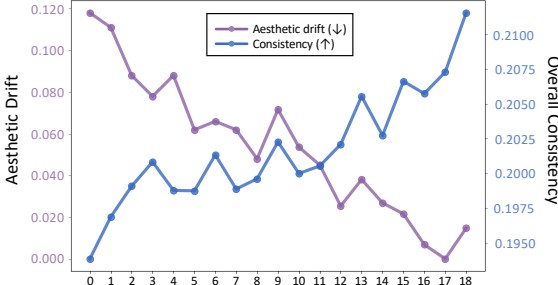

Figure 5. **Ablation study on Deep Sink depth.** We evaluate the effect of attention sink depth on visual quality using Aesthetic Drift ($\downarrow$) and Overall Consistency ($\uparrow$) metrics on 50-second videos from the first 21 prompts in MovieGen (Polyak et al., 2024).

preserves a large portion of the earliest generated tokens in the KV cache to maintain globally informative context, and **Participative Compression**, which selectively retains important tokens in the KV cache while evicting redundant ones. The overall architecture is illustrated in Fig. 3.

### 4.1. Deep Sink

**Attention Analysis.** While recent autoregressive generation models, including Self Forcing (Huang et al., 2025), show promising potential for long-horizon video generation, they still suffer from severe error accumulation when generating videos longer than 5 seconds, which lies beyond their training domain.

Motivated by recent advances in attention sinks for Large Language Models (LLMs) (Xiao et al., 2023), which stabilize attention distributions and improve long-context generation, as illustrated in Fig. 4, we first investigate the attention behavior of pre-trained Self Forcing.

Specifically, we quantify how newly generated frames attend to earlier frames in the KV cache. Contrary to the conventional understanding in LLMs (Xiao et al., 2023) that only a small set of initial KV tokens needs to be retained, our analysis reveals that most attention heads allocate substantial weight not only to the *earliest* frames (e.g., Frames 0–2 in L1H1), but also to frames in the *middle* of the sequence (e.g., Frames 12–14 in L1H1).

**Deepening Attention Sink.** The overall architecture of Deep Sink is presented in Fig. 3(a). Based on our observation, we hypothesize that retaining a larger set of tokens extending into the middle of the sequence is essential for providing global contextual information in long-horizon video generation. To evaluate this hypothesis, we examine the impact of different attention sink sizes on the generation quality of long videos.

To rigorously assess long-horizon generation quality, we adopt two key metrics from VBench (Huang et al., 2024): *Overall Consistency* and *Aesthetic Quality*, which are computed using ViCLIP (Wang et al., 2023) and the LAION

aesthetic predictor (LAION-AI), respectively. Following standard practice in long video generation (Zhang et al., 2026; Yin et al., 2025; Liu et al., 2025), we compute the *Aesthetic Quality Drift*, denoted as $\Delta_{\text{Drift}}^{\text{Quality}}$, defined as the absolute difference in aesthetic quality between the first and last five seconds of each 50-second generated video.

Our results reveal a clear trend (Fig. 5): as the sink frame size increases, *Overall Consistency* consistently improves, while *Aesthetic Quality Drift* ($\Delta_{\text{Drift}}^{\text{Quality}}$) decreases. This finding suggests that intermediate frames serve as crucial visual anchors, effectively preserving visual fidelity throughout long-horizon generation. To the best of our knowledge, this is the first work to systematically investigate attention sink mechanisms during inference in autoregressive video generation.

**Temporal RoPE Adjustment.** RoPE (Su et al., 2024) is widely used as the positional embedding in video diffusion models, and recent architectures (Wan et al., 2025; Teng et al., 2025; Yang et al., 2024) commonly adopt 3D RoPE, which encodes temporal, height, and width dimensions separately. However, attention sinks in video require the model to attend to past frames, and directly applying 3D RoPE under this setting leads to large temporal discrepancies, where tokens at vastly different timestamps (e.g., $t = 1$ vs. $t = 200$) are forced to attend to each other. This breaks the continuity of video and results in (1) flickering, (2) fidelity degradation, and (3) roll-back, where previously sinked frames are regenerated (a detailed analysis is provided in the Appendix A).

To address this, we propose selectively adjusting only the temporal dimensions while preserving the original spatial encoding. Specifically, we selectively modify the temporal RoPE index by applying a temporal offset to the attention sink's temporal index. This reduces the temporal gap between the attention sink and the remaining tokens, while preserving the spatial indices unchanged.

We divide the key and value caches $K$ and $V$ in the current sliding window into two parts: the sink ($K_{\text{sink}}, V_{\text{sink}}$) for deep sink tokens and the tail ($K_{\text{tail}}, V_{\text{tail}}$) for the rest.

$$K = \begin{bmatrix} K_{\text{sink}} \| K_{\text{tail}} \end{bmatrix}, \quad V = \begin{bmatrix} V_{\text{sink}} \| V_{\text{tail}} \end{bmatrix}, \quad (3)$$

where $\begin{bmatrix} \cdot \| \cdot \end{bmatrix}$ denotes concatenation.

Let $s_{\text{tail}}$ denote the first frame index of the tail and let $s_{\text{sink}}$ be the last frame index of the deep sink. We define $\Delta_{\text{sink}}$, which is the temporal gap between $s_{\text{tail}}$ and $s_{\text{sink}}$, as follows:

$$\Delta_{\text{sink}} = s_{\text{tail}} - s_{\text{sink}}. \quad (4)$$

We apply $\Delta_{\text{sink}}$ to $K_{\text{sink}}^{\text{time}}$, which is temporal component of $K_{\text{sink}}$, using the RoPE temporal frequency vector $\boldsymbol{\omega}_t$:

$$K_{\text{sink}}^{\text{time}} \leftarrow K_{\text{sink}}^{\text{time}} \odot \exp\big(i\,\boldsymbol{\omega}_t\,\Delta_{\text{sink}}\big), \quad (5)$$

where $i$ is the imaginary unit, and $\odot$ denotes element-wise

multiplication. This further rotates $K_{\text{sink}}$ to align the relative temporal positions of sink and tail tokens.

## 4.2. Participative Compression

**Motivation** While Deep Sink effectively mitigates fidelity degradation compared to the baseline Self Forcing (Huang et al., 2025), it alone cannot fully alleviate quality degradation in minute-long video generation. When extrapolating from 5-second training clips to sequences more than 12× longer, a critical issue emerges: *degeneration*, where visual fidelity and overall quality progressively deteriorate. This phenomenon is well-documented in autoregressive long-context generation (Holtzman et al., 2019; Ghadia et al., 2025): when generating beyond training length, indiscriminate token retention causes attention to dilute across both relevant and irrelevant context, introducing compounding noise. Beyond its training distribution, the growing KV cache retains increasingly irrelevant tokens, further diluting attention.

Recent analysis of video diffusion models reveals that attention concentrates on a small subset of semantically critical tokens, with the majority contributing minimally to generation (Yang et al., 2025c)—suggesting that pruning low-attention tokens can substantially reduce computation with limited impact on quality. Building on this insight and importance-aware compression (Zhang et al., 2023; Li et al., 2024; Ghadia et al., 2025), we propose an importance-aware KV cache compression method, termed **Participative Compression** (PC), which identifies and retains contextually relevant tokens while pruning those that contribute to attention dilution and error accumulation. As illustrated in Fig. 3(b), PC selectively removes redundant tokens by ranking them according to their aggregated attention scores from recent frames.

**KV Cache Decomposition.** We partition KV cache within the current sliding window into three disjoint subsets:

$$
\begin{aligned}
K &= \begin{bmatrix} K_{\text{sink}} \| K_{\text{cand}} \| K_{\text{rct}} \end{bmatrix}, \\
V &= \begin{bmatrix} V_{\text{sink}} \| V_{\text{cand}} \| V_{\text{rct}} \end{bmatrix},
\end{aligned}
\quad (6)
$$

where $\begin{bmatrix} \cdot \| \cdot \end{bmatrix}$ denotes concatenation.

- **Sink**: $K_{\text{sink}}, V_{\text{sink}}$, containing the earliest $S$ deep sink tokens that are always preserved (Sec. 4.1).
- **Recent**: $K_{\text{rct}}, V_{\text{rct}}$, containing the most recent $R$ tokens, which are excluded from compression to preserve local temporal coherence.
- **Candidate**: $K_{\text{cand}}, V_{\text{cand}}$, containing intermediate tokens between the sink $S$ and recent $R$ tokens, which are subject to compression.

**Importance-Aware Selection.** Given the Budget ($N$), which specifies the target number of tokens to retain after compression, PC is applied when the sliding window

reaches its maximum length of $M$ tokens, compressing the KV cache to a size of $N \leq M$. For each token in $K_{\text{cand}}$ and $V_{\text{cand}}$, PC computes an importance score by aggregating its attention weights across the most recent $R$ tokens. Tokens that are frequently attended to by recent frames are considered more important for maintaining temporal coherence.

PC selects the top $C = N - S - R$ tokens with the highest importance scores to form the compressed set $K_{\text{top}}, V_{\text{top}}$. The final KV cache consists of $N$ tokens in total, comprising $S$ sink, $C$ selected candidate, and $R$ recent tokens.

**Top-$C$ Selection.** To determine which Top-$C$ most relevant tokens to retain, PC computes attention scores between the recent queries $Q_{\text{recent}}$ and candidate keys $K_{\text{cand}}$. We aggregate these scores across all recent queries by summing along the query dimension, producing a importance score $\phi_j$ for each candidate key:

$$\phi_j = \sum_{r=1}^{R} \mathbf{q}_r^{\top} \mathbf{k}_j, \tag{7}$$

where $j$ indexes the candidate keys, $\mathbf{q}_r$ denotes the $r$-th query in $Q_{\text{recent}}$, and $\mathbf{k}_j$ denotes the $j$-th key in $K_{\text{cand}}$. Higher $\phi_j$ indicates higher importance for current generation. We then form the importance vector $\boldsymbol{\phi} = [\phi_1, \phi_2, \ldots, \phi_{|K_{\text{cand}}|}]$ and select Top-$C$ tokens with the highest scores:

$$K_{\text{top}} = \text{Top-C}(\boldsymbol{\phi}). \tag{8}$$

Finally, the compressed cache is formed by concatenating the preserved components in temporal order:

$$K_{\text{comp}} = \left[ K_{\text{sink}} \parallel K_{\text{top}} \parallel K_{\text{rct}} \right], \tag{9}$$

where $K_{\text{sink}}$, $K_{\text{rct}}$ contain keys from the first $S$ and recent $R$, respectively. Values $V_{\text{top}}$ are processed identically. This yields a compact cache structure combining long-term initial context ($S$), selectively important intermediate tokens (Top-$C$), and recent context ($R$), all within fixed budget of $N$.

**Temporal RoPE Unification.** After selecting the Top-$C$ tokens, we apply RoPE adjustment to maintain temporal consistency, following same approach as Deep Sink (Section 4.1). We adjust only the temporal dimension of the Top-$C$ keys' RoPE while preserving spatial information.

Let $s^{\text{top}}$ denote the desired absolute temporal position where the Top-$C$ block should be aligned, and let $s_{\text{base}}^{\text{top}}$ represent the current temporal position of each cached Top-$C$ key. We compute the temporal adjustment:

$$\Delta_{\text{top}} = s^{\text{top}} - s_{\text{base}}^{\text{top}}. \tag{10}$$

This temporal shift $\Delta_{\text{top}}$ is then applied to $K_{\text{top}}^{(\text{time})}$, which is temporal component of $K_{\text{top}}$, re-aligning each Top-$C$ key using the complex phase rotation defined by the RoPE temporal frequencies $\boldsymbol{\omega}_t$:

$$K_{\text{top}}^{\text{time}} \leftarrow K_{\text{top}}^{\text{time}} \odot \exp\left(i\,\boldsymbol{\omega}_t\,\Delta_{\text{top}}\right). \tag{11}$$

where $i$ is the imaginary unit, and $\odot$ denotes element-wise multiplication.

This rotation adjusts the temporal positioning of $K_{\text{top}}$ to create a continuous temporal sequence across all three cache components ($S$, Top-$C$, $R$), preventing temporal discontinuities that would otherwise cause fidelity degradation as demonstrated in Appendix A.

**Efficiency.** Although PC involves gathering and sorting tokens, its computational overhead is minimal since it is only engaged when the sliding window is full and only at the first diffusion timestep ($t = T$). We provide detailed efficiency analysis in Appendix E.

## 5. Experiments

### 5.1. Implementation Details

We use pre-trained Self Forcing (Huang et al., 2025) as our base model. We set sink size $S = 10 \times 1560$ (10 frames), budget $N = 16 \times 1560$ (16 frames), and recent $R = 4 \times 1560$ (4 frames) where 1560 tokens correspond to 1 frame. Additional details are provided in Appendix C.

### 5.2. Evaluation Setting

We compare against baseline autoregressive video diffusion models including CausVid (Yin et al., 2025), Self Forcing (Huang et al., 2025), Rolling Forcing (Liu et al., 2025), and LongLive (Yang et al., 2025b). To evaluate long-horizon visual quality, we use VBench-Long (Huang et al., 2024) with 128 prompts from MovieGen (Polyak et al., 2024), following the same prompt selection protocol as Self Forcing++ (Cui et al., 2025). Following Self Forcing (Huang et al., 2025), each prompt is refined using Qwen2.5-7B-Instruct (Yang et al., 2025a). We further conduct a user study to evaluate color consistency, dynamic motion, subject consistency, and overall quality. Finally, we evaluate visual stability using the state-of-the-art Vision Language Model (VLM) Gemini 2.5-Pro (Comanici et al., 2025), following the protocol of Self Forcing++ (Cui et al., 2025).

### 5.3. Comparison

**Quantitative Results.** Table 1 summarizes the quantitative comparison. Although Deep Forcing is a training-free method built upon pre-trained Self Forcing, it achieves performance comparable to training-based approaches (Yang et al., 2025b; Liu et al., 2025), which are distilled or explicitly trained for long-horizon video generation on top of Self Forcing. Specifically, we achieve better performance in overall consistency and imaging quality compared to LongLive (Yang et al., 2025b), and better aesthetic quality than Rolling Forcing (Liu et al., 2025).

*Table 1.* **Quantitative comparison on long-horizon video generation.** We evaluate Deep Forcing against open-source autoregressive video diffusion baselines on 30-second and 60-second videos using multiple quality metrics from VBench-Long (Huang et al., 2024).

| Model | Throughput (FPS) ↑ | Dynamic Degree ↑ | Motion Smoothness ↑ | Overall Consistency ↑ | Imaging Quality ↑ | Aesthetic Quality ↑ | Subject Consistency ↑ | Background Consistency ↑ |
|---|---|---|---|---|---|---|---|---|
| *Trained with Attention Sink* | | | | **30 seconds** | | | | |
| Rolling Forcing | 15.79 | 30.71 | 98.75 | 20.99 | 70.58 | 60.24 | 98.12 | 96.91 |
| LongLive | 18.16 | 45.55 | 98.76 | 20.16 | 69.07 | 61.51 | 97.97 | 96.83 |
| *Trained **without** Attention Sink* | | | | | | | | |
| CausVid | **15.78** | 47.21 | 98.08 | 19.15 | 66.36 | 59.77 | **97.92** | **96.77** |
| Self Forcing | **15.78** | 36.62 | **98.63** | 20.50 | 68.58 | 59.44 | 97.34 | 96.47 |
| **Deep Forcing (Ours)** | 15.75 | **57.56** | 98.27 | **20.54** | **69.31** | **60.68** | 97.34 | 96.48 |
| *Trained with Attention Sink* | | | | **60 seconds** | | | | |
| Rolling Forcing | 15.79 | 31.35 | 98.69 | 20.64 | 70.25 | 59.75 | 97.97 | 96.76 |
| LongLive | 18.16 | 43.49 | 98.75 | 20.29 | 69.11 | 61.29 | 97.85 | 96.74 |
| *Trained **without** Attention Sink* | | | | | | | | |
| CausVid | **15.78** | 46.44 | 98.09 | 18.78 | 65.84 | 59.42 | **97.81** | **96.75** |
| Self Forcing | **15.78** | 31.98 | 98.21 | 18.63 | 66.33 | 56.45 | 96.82 | 96.31 |
| **Deep Forcing (Ours)** | 15.75 | **57.19** | **98.23** | **20.38** | **69.27** | **59.86** | 96.96 | 96.32 |

*Table 2.* **Ablation study of our components.** Effects of Deep Sink (DS) and Participative Compression (PC) on top of Self Forcing (SF).

| Method | Dynamic Degree ↑ | Motion Smoothness ↑ | Overall Consistency ↑ | Imaging Quality ↑ | Aesthetic Quality ↑ | Subject Consistency ↑ | Background Consistency ↑ |
|---|---|---|---|---|---|---|---|
| SF (Baseline) | 36.62 | **98.63** | 20.50 | 68.58 | 59.44 | **97.34** | 96.47 |
| SF + DS | 48.58 | 98.18 | **20.54** | 68.62 | 59.99 | 97.25 | 96.43 |
| SF + DS + PC (**Ours**) | **57.56** | 98.27 | **20.54** | **69.31** | **60.68** | **97.34** | **96.48** |

Notably, our method excels in dynamic degree, producing more dynamic motions than trained methods (Liu et al., 2025; Yang et al., 2025b), despite not being optimized for this aspect. We attribute this to our training-free approach, which avoids the potential motion constraints introduced when models are trained to anchor with attention sinks.

**Qualitative Results.** The qualitative results in Figure 7 demonstrate strong visual quality, with our training-free method producing high-quality frames comparable to or better than training-based baselines. Notably, our videos exhibit more dynamic motion in both camera and subject movement, yielding more visually expressive results compared to existing approaches. Additional qualitative results are provided in Appendix R.

**User Study.** We conducted a user study with 24 participants following the Two-Alternative Forced Choice (2AFC) protocol, evaluating color consistency, dynamic motion, subject consistency, and overall quality. As shown in Table 4, participants demonstrated a clear preference for Deep Forcing across all aspects, particularly in subject consistency—corroborating that perceptual quality remains high despite lower VBench-Long (Huang et al., 2024) subject consistency scores. See Appendix S for detail.

**VLM evaluation.** We further evaluate visual stability using Gemini 2.5-Pro (Comanici et al., 2025), following the protocol of Self Forcing++ (Cui et al., 2025). We use the same prompt to ask the VLM to score each 30-second video in terms of exposure stability and degradation. Then we normalize the resulting scores to 100. As shown in Tab. 3, our training-free method achieves visual stability comparable to

*Table 3.* **Visual stability compared with the baselines using Gemini 2.5-Pro.** Methods are additionally categorized by whether they are trained with an attention sink.

| Method | Attention Sink Training | Visual Stability |
|---|---|---|
| CausVid (Yin et al., 2025) | No | 42.84 |
| Self Forcing (Huang et al., 2025) | No | 43.94 |
| Rolling Forcing (Liu et al., 2025) | Yes | 72.60 |
| LongLive (Yang et al., 2025b) | Yes | 78.58 |
| **Deep Forcing (Ours)** | No | 75.44 |

*Table 4.* **User study results.** Values indicate the percentage of votes favoring Deep Forcing over baseline methods.

| Method | Color Cons. | Dyn. Motion | Subject Cons. | Overall Quality |
|---|---|---|---|---|
| CausVid | 98.9% | 95.8% | 96.8% | 100% |
| Self Forcing | 85.9% | 86.9% | 84.8% | 87.9% |
| LongLive | 71.2% | 83.5% | 72.2% | 72.2% |
| Rolling Forcing | 76.7% | 76.7% | 80.0% | 78.9% |

that of recent methods (Liu et al., 2025; Yang et al., 2025b).

### 5.4. Ablation Study

We conducted ablation studies to evaluate the contributions of each components. We measure relevant VBench-Long metrics on 30 second videos.

**Effect of Deep Sink & Participative Compression.** We evaluate three variants: naive Self Forcing (Huang et al., 2025), Self Forcing with only Deep Sink with sink length $S$ = 10 frames, and Self Forcing with both Deep Sink ($S$ = 10 frames) and Participative Compression ($N$ = 16, $R$ = 4). As shown in Table 2, Deep Forcing demonstrates progressive improvements in dynamic degree, overall consistency, and

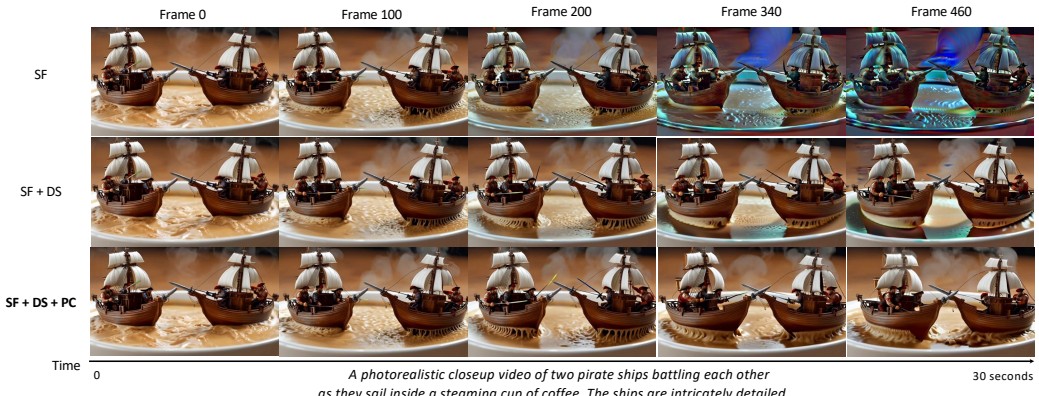

*Figure 6.* **Qualitative ablation results over 30-second generation:** Comparison of **Self Forcing (SF)** (Huang et al., 2025), SF with Deep Sink (SF+DS), and SF with both Deep Sink and Participative Compression (**Deep Forcing**). Baseline SF exhibits severe color drift. SF+DS improves stability but shows residual artifacts. Deep Forcing maintains consistent visual quality.

image quality as components are added. Notably, dynamic degree improves substantially through the Deep Forcing framework, enabling the generation of significantly more dynamic scenes compared to baseline methods.

### 5.5. Ablation Visualization

Figure 6 visualizes our ablation study results. When generating long videos with Self Forcing (SF) alone (top row), error accumulation leads to severe fidelity degradation and visual quality deteriorates as colors drift toward over-saturation. Adding Deep Sink (SF + DS) substantially reduces fidelity degradation and maintains more consistent colors. However, subtle artifacts persist at frame 460, including slight color shift in the coffee and texture blur in the ship details. When both Deep Sink and Participative Compression are applied (**Deep Forcing**), noticeable degradation is effectively eliminated. This validates that our complete framework successfully mitigates long-horizon error accumulation while preserving both overall visual quality and fine-grained details.

### 5.6. Applications

Our method generalizes to other Self Forcing-based models, including world models such as Matrix-Game 2.0 (He et al., 2025) (Fig. 8; Appendix G). Additionally, Deep Forcing supports interactive prompting, enabling users to modify prompts during generation to steer video content (Fig. 9; Appendix F).

### 6. Conclusion

We introduced Deep Forcing, a training-free approach for autoregressive long video generation that mitigates error accumulation through **Deep Sink** and **Participative Compression**. Our method enables long video generation while preserving visual fidelity and motion dynamics, achieving comparable performance to training-based methods.

### Impact Statement

The objective of our work is to advance training-free long-horizon autoregressive video generation by mitigating error accumulation through KV-cache management in video diffusion models. By making minute-long, persistent, and interactive video generation more efficient without additional training, Deep Forcing may broaden access to such systems. Potential societal consequences are therefore similar to those of other generative video and interactive world models, including possible misuse for misleading or unlabeled synthetic media. We do not anticipate impacts beyond those already well established for generative video models.

### Acknowledgements

This research was supported by Institute of Infor-mation & communications Technology Planning & Evaluation (IITP) grant funded by the Korea government (MSIT) (RS-2019-II190075, RS-2024-00509279, RS-2025-II212068, RS-2023-00227592, RS-2025- 02214479, RS-2024-00457882, RS-2025-25441838, RS-2025-25441838, RS-2025-02214479, RS-2025-02217259, RS-2026-25519202) and the Culture, Sports, and Tourism R&D Program through the Korea Creative Content Agency grant funded by the Ministry of Culture, Sports and Tourism (RS-2024-00345025, RS-2024-00333068, RS-2023-00222280, RS-2023-00266509), and National Research Foundation of Korea (RS-2024-00346597). This research was supported by the "Advanced GPU Utilization Support Program" funded by the Government of the Republic of Korea (Ministry of Science and ICT). This research was supported by the AI Computing Infrastructure Enhancement (GPU Rental Support) User Support Program funded by the Ministry of Science and ICT (MSIT), Republic of Korea (RQT-25-120217).

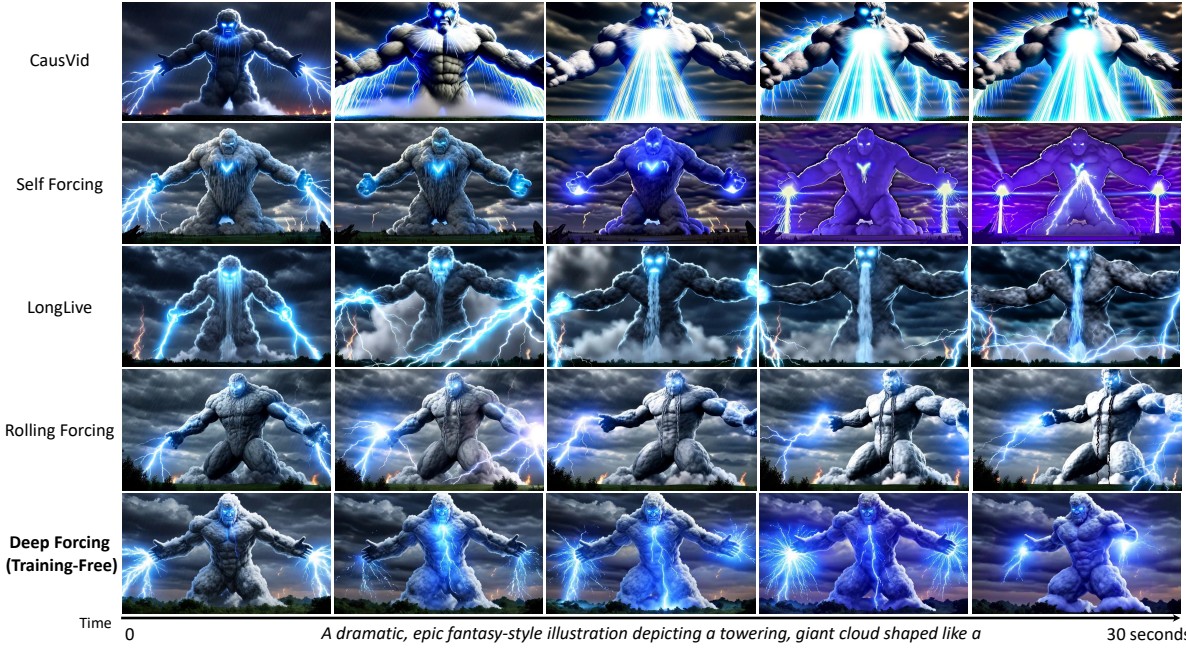

*Figure 7.* **Qualitative result on a 30-second video.** Frame-by-frame comparison across different methods for a same prompt. Deep Forcing (training-free) achieves temporal consistency and visual quality comparable to training-based baselines (CausVid (Yin et al., 2025), Self Forcing (Huang et al., 2025), LongLive (Yang et al., 2025b), Rolling Forcing (Liu et al., 2025)) while generating more dynamic content with greater subject consistency.

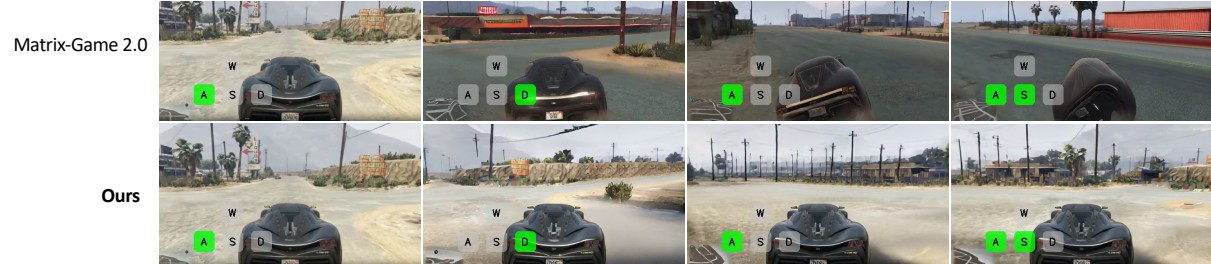

*Figure 8.* **Deep Forcing on World Model.** We apply our training-free method to Matrix-Game 2.0 (He et al., 2025), an open-source Self Forcing-based world model. Over a 1 minute rollout, vanilla Matrix-Game 2.0 (top row) exhibits noticeable appearance drift(e.g., color,style, distortion). Deep Forcing (bottom row) substantially reduces such drift, preserving the input image's color, style.

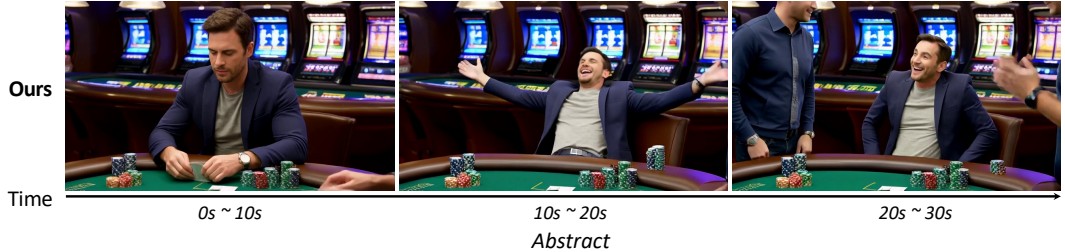

*Abstract*

*0s ~ 10s: A male player in his late 30s with short dark hair sits at a well-lit poker table and tightly grips his hole cards, wearing a tense, serious expression.*

*10s ~ 20s: The same player flicks his cards onto the felt, then leans back in the chair with arms spread wide in celebration.*

*20s ~ 30s: The same player reveals the winning hand and leans back in celebration while a nearby patron claps and cheers for the winner.*

*Figure 9.* **Interactive Prompting.** Deep Forcing enables users to modify prompts during generation, allowing real-time interactive control over video content.

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

# Appendix

In this appendix, we provide comprehensive details, including:

- Comparison with other sink mechanisms (Yang et al., 2025b; Liu et al., 2025) (Appendix A)
- Qualitative results on different sink sizes (Appendix B)
- Participative Compression details (Appendix C)
- Denoising query is not just noise (Appendix D)
- FPS measurements (Appendix E)
- Interactive prompting (Appendix F)
- Deep Forcing on World Model (Appendix G)
- Difference with FramePack (Appendix H)
- Design Justification of Deep Forcing (Appendix I)
- Robustness to Hyperparameter Choice (Appendix J)
- Transferability to Causal Forcing (Appendix K)
- Ablation on Per-Timestep Top-$C$ Selection (Appendix L)
- Justification of Attention-Based Token Selection (Appendix M)
- Additional Quantitative Evaluation (Appendix N)
- VLM-based Verification of Motion Quality (Appendix O)
- Detailed Comparison with LLM KV Cache Methods (Appendix P)
- Inter-Rater Agreement in User Study (Appendix Q)
- More qualitative results (Appendix R)
- User study protocol (Appendix S)
- Additional attention visualization (Appendix T)
- Limitations and Future Works. (Appendix U)

## A. The Tale of Three Sinks

Concurrent works (Yang et al., 2025b; Liu et al., 2025) propose different attention sink mechanisms for Self Forcing-like architectures, typically with model training or distillation. In this section, we compare these approaches in a training-free setting to evaluate their effectiveness when applied directly to pretrained Self Forcing model.

Specifically, we compare three attention sink strategies: LongLive-style (Yang et al., 2025b), Rolling Forcing-style (Liu et al., 2025), and Ours. LongLive-style applies attention sinks to the first **3 frames** without RoPE adjustment. Rolling Forcing-style also uses **3 frame** sinks but incorporates (1) storing raw keys and (2) dynamically re-applying RoPE when rolling occurs. Qualitative comparisons are presented in Figure 10.

Figure 10 shows that our method achieves substantially better results. The LongLive-style attention sink, which does not adjust RoPE, exhibits progressive failure modes: **fidelity degradation** appears at frame 800, followed by **flickering** artifacts at frame 801, and culminating in **roll-back** behavior at frame 802 where the generation reverts to early sinked frames. These issues also occur in LongLive (Yang et al., 2025b), which was explicitly trained on long videos using this attention sink mechanism (Fig. 1).

Although Rolling Forcing-style attention sink (Liu et al., 2025) employs Dynamic RoPE, which reapplies positional encodings to the entire cached key set, it still exhibits severe fidelity degradation at frames 800–801.

This comparison demonstrates that both deep sink and RoPE adjustment are essential for long video generation.

Note that this is a training-free ablation on a fixed pretrained Self Forcing backbone — not a comparison against the fully trained LongLive and Rolling Forcing models.

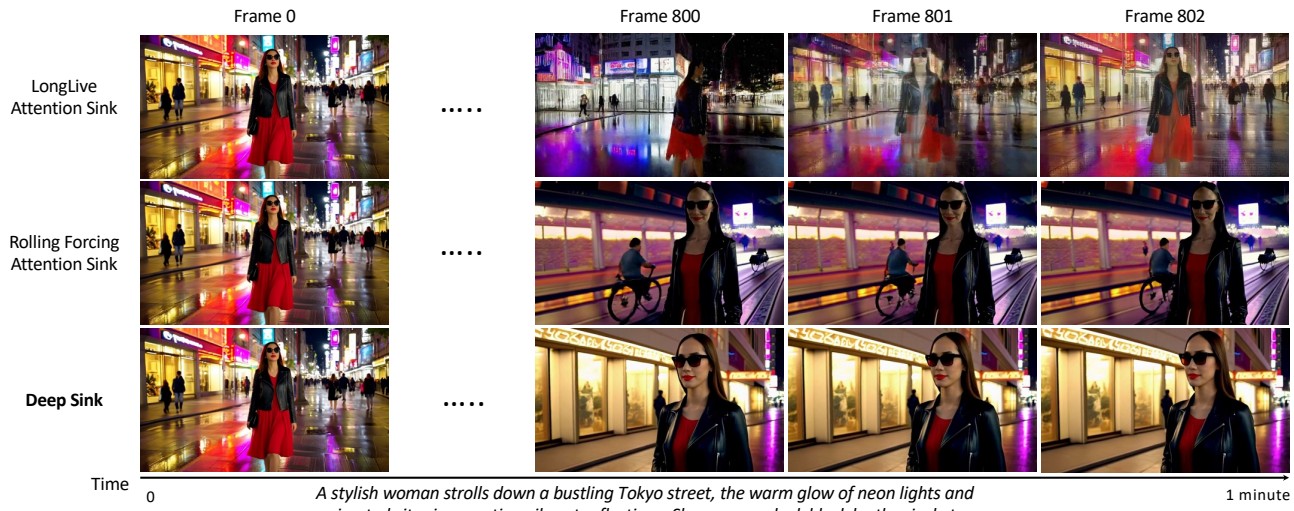

*Figure 10.* **Qualitative results on different Attention Sink.** The result shows that Deep Sink substantially outperforms both LongLive-style (Yang et al., 2025b) and Rolling Forcing-style (Liu et al., 2025) attention sinks.

## B. Qualitative Results on Different Sink Size

Figure 11 presents qualitative comparisons across sink sizes ranging from 0 to 18 frames, as analyzed in Section 4.1. We evaluate two diverse prompts on 60-second generation.

With no attention sink (Sink 0), severe fidelity degradation emerges rapidly, where the monster's texture deteriorates and colors shift noticeably by frame 230, with complete quality collapse by frame 690. Similarly, the SUV scene exhibits significant fidelity degradation. As the sink size increases to 4 and 9 frames, degradation is progressively reduced but remains visible in fine details.

Once the sink size exceeds 10 frames (Sink 14), fidelity degradation is substantially reduced. However, excessively large sinks (Sink 18) exhibit repetitive generation where early frames are over-preserved.

These results validate our optimal sink range of 10–15 frames (40–60% of the sliding window). While Deep Sink substantially mitigates degradation, it alone proves insufficient to maintain visual fidelity throughout minute-long generation across diverse scenes, as demonstrated in our extended evaluations (Section 5.4).

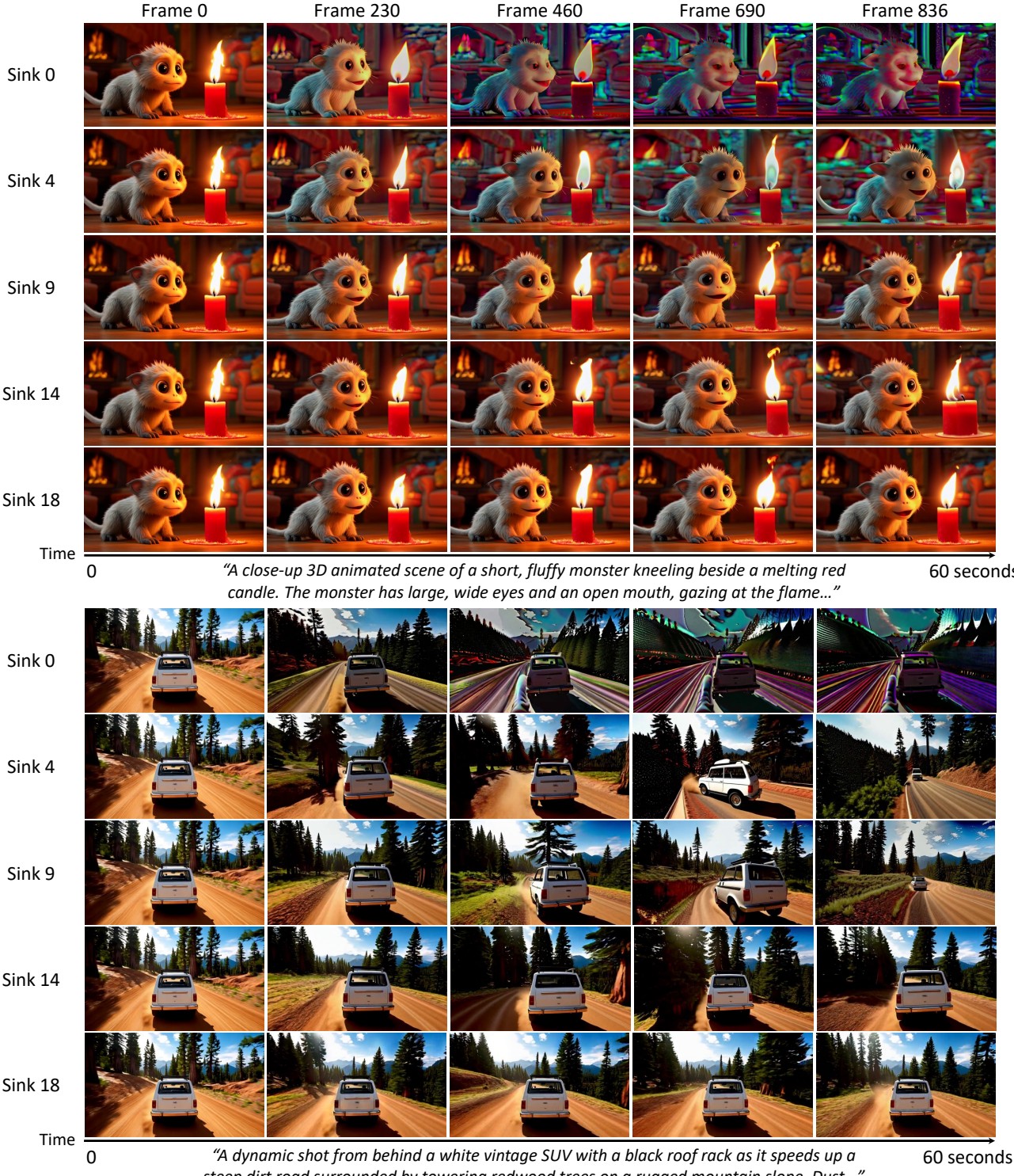

*Figure 11.* **Qualitative comparison of different sink sizes on 60-second videos.** As the sink size decreases, degradation becomes more severe. Once the sink size exceeds 10 frames, degradation is substantially reduced.

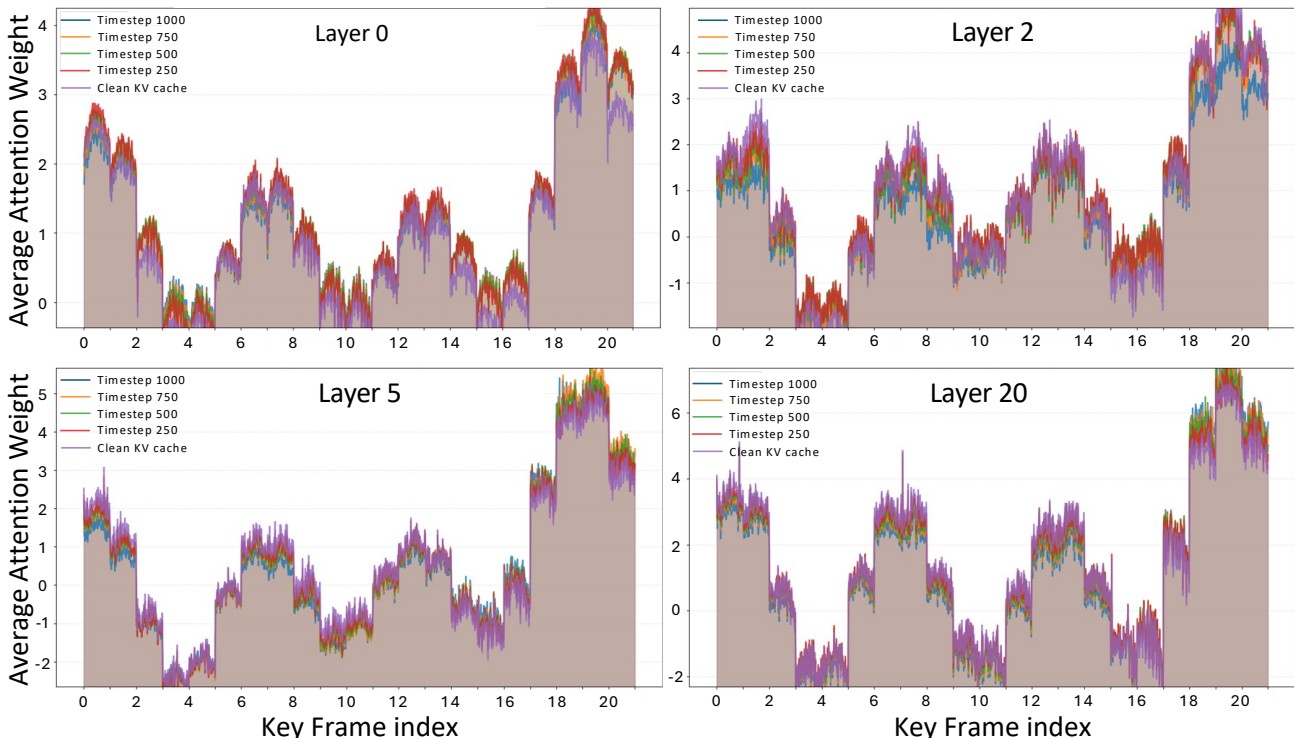

*Figure 12.* **Attention weight consistency across diffusion timesteps.** Query-averaged attention weights showing how each key frame is attended when generating the last chunk (frames 19–21) at different denoising timesteps. The consistent attention patterns across timesteps (1000, 750, 500, 250, and the final clean KV cache) demonstrate that Top-$C$ tokens selected at the initial timestep ($t = 1000$) remain valid and contextually relevant throughout the entire denoising process.

## C. Participative Compression Details

In this section, we provide additional details and analysis of Participative Compression (PC) beyond what was presented in Section 4.2.

Each layer maintains its own KV cache, which undergoes compression as follows:

$$
\begin{aligned}
\left[\, K_{\text{sink}} \parallel K_{\text{cand}} \parallel K_{\text{rct}} \,\right] &\rightarrow \left[\, K_{\text{sink}} \parallel K_{\text{top}} \parallel K_{\text{rct}} \,\right], \\
\left[\, V_{\text{sink}} \parallel V_{\text{cand}} \parallel V_{\text{rct}} \,\right] &\rightarrow \left[\, V_{\text{sink}} \parallel V_{\text{top}} \parallel V_{\text{rct}} \,\right].
\end{aligned}
\tag{12}
$$

PC compresses only the intermediate tokens between the sink and recent. This design preserves both the initial context ($K_{\text{sink}}, V_{\text{sink}}$) and recent context ($K_{\text{rct}}, V_{\text{rct}}$) without modification, while compressing only the candidate tokens ($K_{\text{cand}}, V_{\text{cand}}$) to retain the most important visual and contextual information ($K_{\text{top}}, V_{\text{top}}$).

This compression occurs independently in each layer. Importantly, PC is applied only at the first diffusion timestep ($t = 1000$) when the cache reaches over its maximum window length. The tokens selected at this initial timestep remain fixed throughout subsequent denoising steps.

Figure 12 validates this design choice by demonstrating that attention patterns remain consistent across timesteps. The tokens deemed important when generating frames 19–21 exhibit similar importance scores across different diffusion timesteps ($t = 1000, 750, 500, 250$), confirming that the Top-$C$ selection at $t = 1000$ captures tokens that remain contextually relevant throughout the entire denoising process.

**Algorithm** Deep Forcing inference is illustrated in Alg. 1. When the KV cache reaches the window limit, PC is triggered only at the first diffusion step ($t = T$). We preserve the first $S$ sink tokens and the most recent $R$ tokens, treating the remaining middle tokens as candidates. We score candidate tokens by aggregating attention from the recent queries, retain the top $C = N - R - S$ tokens, apply temporal RoPE unification to the retained keys to maintain temporal coherence, and rebuild the cache in temporal order as $[K_{\text{sink}} \parallel K_{\text{top}} \parallel K_{\text{rct}}]$ (and same process for $V$).

---

**Algorithm 1** Participative Compression with Deep Sink

---

1: **Input:** KV cache $[K, V]$ of size $M$; Sink size $S$; Recent $R$; Top-C capacity $C$; timestep $t$; first timestep $T$
2: **if** $M \geq \text{MAX\_WINDOW\_LENGTH}$ **and** $t = T$ **then**
3:     *// Partition cache into three regions*
4:     $\mathcal{I}_{\text{sink}} \leftarrow [0, S)$                                                  *// first $S$ tokens*
5:     $\mathcal{I}_{\text{rct}} \leftarrow [M - R, M)$                                           *// last $R$ tokens*
6:     $\mathcal{I}_{\text{cand}} \leftarrow [S, M - R)$                                       *// candidate tokens*
7:     **if** $|\mathcal{I}_{\text{cand}}| > 0$ **and** $C > 0$ **then**
8:         *// Compute importance scores (Eq. 7)*
9:         $Q_{\text{rct}} \leftarrow Q[\mathcal{I}_{\text{rct}}]$                                     *// recent queries*
10:        $K_{\text{cand}} \leftarrow K[\mathcal{I}_{\text{cand}}]$                                 *// candidate keys*
11:        **for** $j = 1$ **to** $|\mathcal{I}_{\text{cand}}|$ **do**
12:           $\phi_j \leftarrow \sum_{r=1}^{R} \mathbf{q}_r^\top \mathbf{k}_j$                            *// aggregate attention*
13:        **end for**
14:        *// Select top-C tokens (Eq. 8)*
15:        $\boldsymbol{\phi} \leftarrow [\phi_1, \phi_2, \ldots, \phi_{|\mathcal{I}_{\text{cand}}|}]$
16:        $\mathcal{I}_{\text{top}} \leftarrow \text{TOPC}(\boldsymbol{\phi})$                                *// select $C$ highest*
17:        *// Temporal RoPE unification (Sec. 4.2)*
18:        $\Delta_{\text{top}} \leftarrow s^{\text{top}} - s_{\text{base}}^{\text{top}}$
19:        $K_{\text{top}}^{(\text{time})} \leftarrow K_{\text{top}}^{(\text{time})} \odot \exp(i\boldsymbol{\omega}_t \Delta_{\text{top}})$
20:     **else**
21:        $\mathcal{I}_{\text{top}} \leftarrow \varnothing$
22:     **end if**
23:     *// Assemble compressed cache (Eq. 9)*
24:     $K_{\text{comp}} \leftarrow [K_{\text{sink}} \| K_{\text{top}} \| K_{\text{rct}}]$
25:     $V_{\text{comp}} \leftarrow [V_{\text{sink}} \| V_{\text{top}} \| V_{\text{rct}}]$
26:     **return** $K_{\text{comp}}, V_{\text{comp}}$
27: **else**
28:     **return** $K, V$                                                         *// no compression*
29: **end if**
30: **Output:** Compressed $(K_{\text{comp}}, V_{\text{comp}})$ or original $(K, V)$

---

**Participative Compression Ablation.** As shown in Figure 3 in the main paper, Participative Compression (PC) can leverage both current denoising query tokens and clean past query tokens from previously generated frames to select the Top-$C$ candidates. We evaluate the effect of using each type independently versus combining them together.

Table 5 compares these three strategies. When Top-$C$ is selected using only clean past tokens (Only Past), the method achieves an image quality of 68.54 and overall consistency of 20.47. When selection relies solely on currently denoising tokens (Only Denoising), the noisy nature of these queries at the initial timestep ($t = 1000$) leads to slightly lower image quality (68.24) and motion smoothness (97.86), likely due to unstable token selection at the initial denoising step. Combining both query types (Both) achieves the highest scores across all metrics, including motion smoothness, overall consistency, and image quality. The clean past queries appear to provide relatively stable importance estimates, while the current denoising queries help ensure the selected tokens remain relevant to the immediate generation context, suggesting complementary benefits from their combination.

**Participative Compression Hyper-parameter Ablation.** To determine the optimal budget $N$ and recent $R$, we conduct an ablation study on 30-second videos using VBench-Long (Huang et al., 2024) (Table 6). While several configurations achieve higher scores on individual metrics, we chose the Budget/Recent ratio ($N/R$) of **16/4** (i.e., $N = 16 \times 1560$, $R = 4 \times 1560$, where 1560 tokens denote 1 frame) as the optimal setting, as it provides the most balanced performance across VBench-Long dimensions (e.g., overall consistency, imaging quality, and aesthetic quality).

**Top-C Visualization.** Figure 13 visualizes a subset of Top-$C$ tokens selected during the first rolling step, spatially aligned to their positions in Frame 37 when generating Frame 82. The yellow highlighted regions indicate the spatial positions of tokens selected as Top-$C$ within the frame. These highlighted regions reveal semantic alignment with contextually important

*Table 5.* **Ablation on Participative Compression.**

| Method | Dynamic Degree ↑ | Motion Smoothness ↑ | Overall Consistency ↑ | Imaging Quality ↑ | Aesthetic Quality ↑ | Subject Consistency ↑ | Background Consistency ↑ |
|---|---|---|---|---|---|---|---|
| Only Denoising | **59.17** | 97.86 | 20.44 | 68.24 | 60.36 | 97.04 | 96.26 |
| Only Past | 58.40 | 97.91 | 20.47 | 68.54 | 60.58 | **97.42** | 96.25 |
| Both | 57.56 | **98.27** | **20.54** | **69.31** | **60.68** | 97.34 | **96.48** |

*Table 6.* **Ablation study on Participative Compression hyperparameters.** We evaluate different Budget/Recent ($N/R$) configurations on 30-second videos using VBench-Long (Huang et al., 2024).

| Model(N/R) | Dynamic Degree ↑ | Motion Smoothness ↑ | Overall Consistency ↑ | Imaging Quality ↑ | Aesthetic Quality ↑ | Subject Consistency ↑ | Background Consistency ↑ |
|---|---|---|---|---|---|---|---|
| Ours(15/4) | 56.88 | 98.22 | 20.47 | 69.27 | 60.38 | **97.36** | **96.51** |
| Ours(16/4) | 57.56 | 98.27 | 20.54 | **69.31** | **60.68** | 97.34 | 96.48 |
| Ours(16/5) | 53.28 | **98.35** | **20.55** | 69.29 | 60.61 | 97.21 | 96.45 |
| Ours(17/4) | 58.01 | 98.30 | 20.44 | 69.13 | 60.35 | 97.27 | 96.39 |
| Ours(17/5) | 55.90 | 98.32 | 20.48 | 69.18 | 60.55 | 97.32 | 96.39 |
| Ours(17/6) | 57.11 | 98.21 | 20.36 | 69.23 | 60.41 | 96.91 | 96.33 |
| Ours(18/4) | **58.45** | 97.98 | 20.27 | 68.93 | 60.17 | 97.09 | 96.42 |

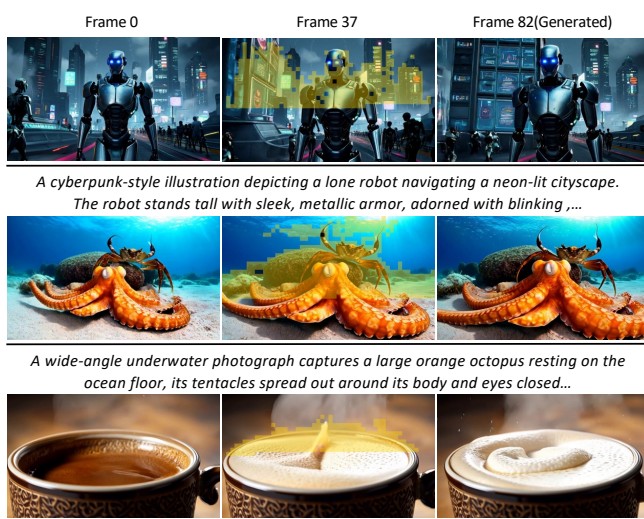

Frame 0     Frame 37     Frame 82(Generated)

*A cyberpunk-style illustration depicting a lone robot navigating a neon-lit cityscape. The robot stands tall with sleek, metallic armor, adorned with blinking ,…*

*A wide-angle underwater photograph captures a large orange octopus resting on the ocean floor, its tentacles spread out around its body and eyes closed…*

*A macro shot of a volcanic eruption in a coffee cup, capturing the dramatic moment in vivid detail. The coffee cup is filled with rich,…*

*Figure 13.* **Visualization of Top-$C$ token selection.** For each example, Frame 37 **(middle)** shows the Top-$C$ tokens selected for generating Frame 82 **(right)**. Yellow highlights indicate the spatial locations of tokens chosen as Top-$C$. Our method effectively identifies and preserves regions that are critical for maintaining contextual coherence during subsequent generation.

content: the robot's body and background architecture, the octopus tentacles and crab, and the circular coffee cup structure. This demonstrates that our method identifies and retains semantically salient regions critical for maintaining contextual coherence in subsequent generation.

## D. Denoising Query Is Not Just Random Noise

While the denoising queries at the initial timestep ($t = 1000$) are inherently noisy, they may still carry meaningful signal for identifying important tokens. Figure 12 suggests this by showing consistent attention patterns across timesteps, but to more conclusively demonstrate the effectiveness of noisy queries, we directly compare Top-$C$ selection based on denoising queries versus Gaussian random selection. For denoising query-based selection, we compute $QK^T$ using only the currently denoising query tokens, then select the Top-$C$ candidates. For Gaussian random selection, we assign each candidate token a score sampled from $\mathcal{N}(0, 1)$ and select the Top-$C$ based on these random scores.

Figure 14 illustrates the stark difference. Random selection exhibits severe scene repetition and context loss, as randomly chosen anchors fail to preserve coherent contextual information. In contrast, denoising query-based selection generates context-aware videos with notably better subject consistency and context.

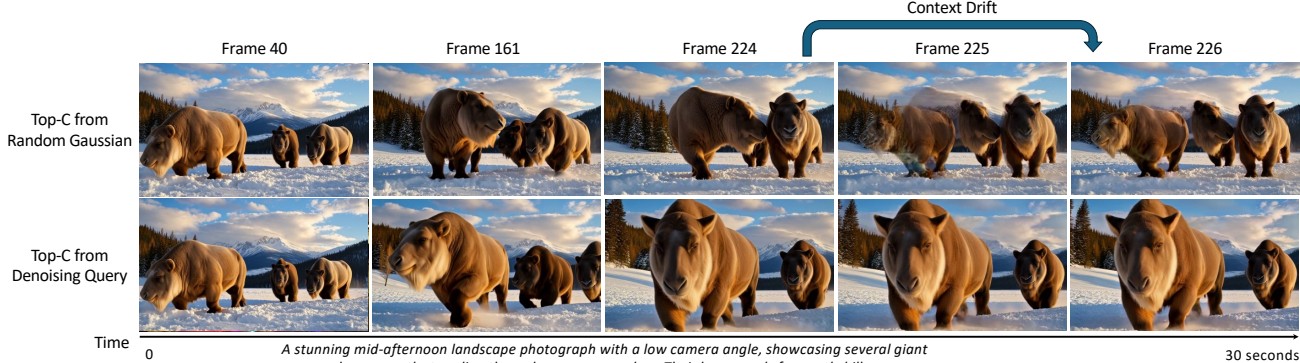

*Figure 14.* **Qualitative comparison: Random Top-$C$ vs. Denoising Query Top-$C$.** Gaussian random selection causes severe artifacts during compression - faces abruptly rotate, heads appear floating in mid-air, and random context drift occurs, resulting in incoherent scene transitions. In contrast, denoising query-based selection maintains subject consistency with natural emergent camera movements and preserves contextual coherence throughout the generation.

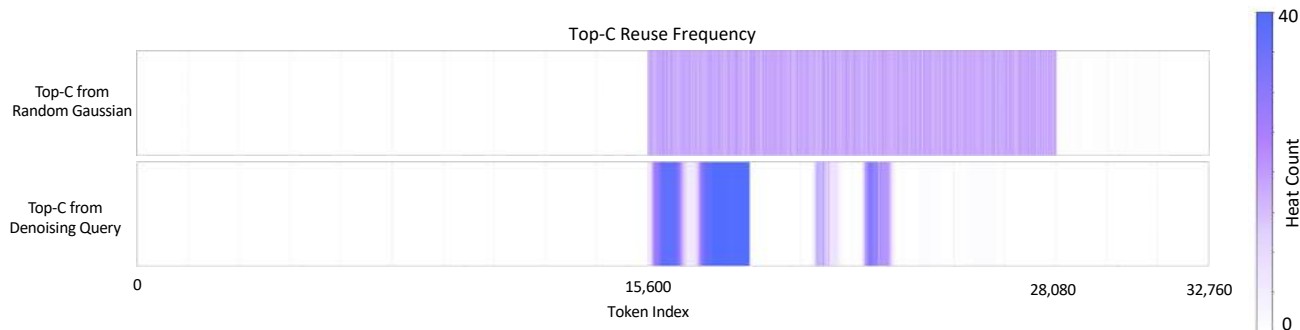

*Figure 15.* **Token-wise Top-$C$ selection frequency heatmap during 1-minute generation.** Color intensity ranges from white (rarely selected) to dark purple (frequently selected as Top-$C$), indicating how often each token is reused throughout the generation. The x-axis spans tokens 0–32,760, where 0–15,600 are Deep Sink tokens, 15,600–28,080 are candidates for compression, and 28,080+ are recent tokens. Gaussian random selection (top) distributes selections uniformly across candidate tokens, whereas denoising query-based selection (bottom) concentrates heavily on specific semantically important tokens—particularly those immediately after the sink boundary—that effectively bridge established and newly formed context.

Figure 15 further validates this through token selection frequency heatmaps over 1-minute generation, where color intensity (white to dark purple) indicates selection frequency. The x-axis spans tokens 0–32,760: tokens 0–15,600 are Deep Sink, 15,600–28,080 are compression candidates, and 28,080+ are recent. Gaussian random selection (top) distributes uniformly across candidates, while denoising query-based selection (bottom) concentrates heavily on specific positions, particularly immediately after the sink boundary (15,600).

Notably, these high-frequency positions do not correspond to fixed frames, as tokens are evicted during compression, subsequent tokens shift into these slots. The concentration at positions near 15,600 indicates that these positional slots are consistently selected regardless of frame identity, as they bridge established context (sink) and current generation (recent), serving as semantically important anchors. This positional selectivity demonstrates meaningful contextual relationships rather than arbitrary noise.

We hypothesize this effectiveness stems from: (1) Self Forcing's 4-step distilled diffusion enabling rapid convergence to meaningful attention patterns despite noisy queries at $t = 1000$, and (2) per-layer KV caching allowing independent selection of semantically important tokens based on layer-specific contextual relevance.

## E. FPS Measurements

We evaluate inference throughput on a single NVIDIA H100 GPU when generating 60-second videos. Table 7 demonstrates that Deep Forcing maintains throughput comparable to baseline Self Forcing, achieving 15.75 FPS versus 15.78 FPS. Despite the computational overhead of compression, our method balances two competing factors: (1) compressing from 21 to 16 frames requires additional computation, but (2) generating subsequent frames with only 16 cached frames incurs lower

*Table 7.* **Throughput Comparison on a single H100 GPU. Latency is measured after first rolling.**

| Method | FPS | Latency(Min/Max) |
|---|---|---|
| Self Forcing (Huang et al., 2025) | 15.78 | 0.770 / 0.776s |
| **Deep Forcing (Ours)** | 15.75 | 0.747 / 0.797s |

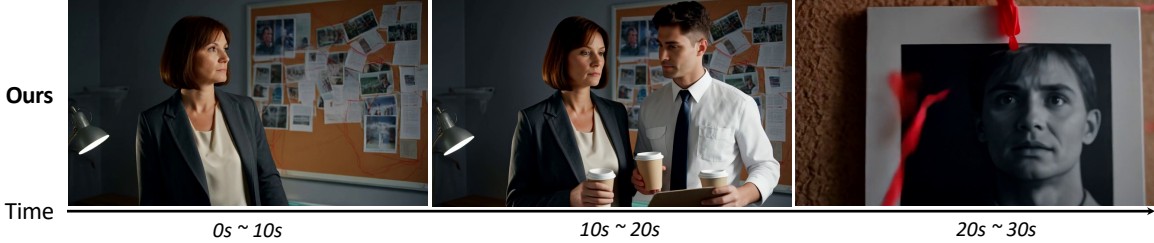

**Ours**

Time

| 0s ~ 10s | 10s ~ 20s | 20s ~ 30s |

*Abstract*

*0s ~ 10s: A female detective in her 40s stares at a large corkboard covered with photos and notes connected by red string in her dimly lit office.*

*10s ~ 20s: Her partner, a younger male detective in his early 30s, enters the office holding two steaming Styrofoam cups of coffee, ready for another long night.*

*20s ~ 30s: The camera zooms in on the corkboard and settles on a black-and-white photograph of a key witness, their face etched with worry, a red string pinned to the photo.*

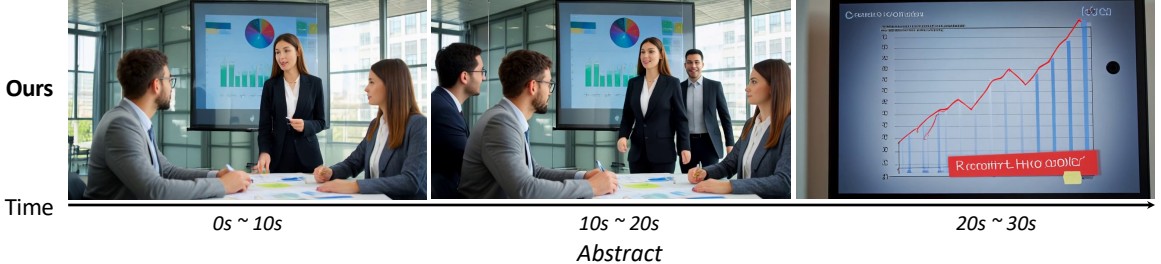

**Ours**

Time

| 0s ~ 10s | 10s ~ 20s | 20s ~ 30s |

*Abstract*

*0s ~ 10s: A young executive in a business suit gives a presentation in a modern, glass-walled boardroom, pointing to a chart on a large screen as colleagues watch attentively.*

*10s ~ 20s: The presentation is interrupted when the company's charismatic CEO unexpectedly enters the room with a broad smile.*

*20s ~ 30s: The camera slowly zooms in on the presentation screen and settles on a close-up of a rising line chart.*

*Figure 16.* **Interactive prompting** Deep Forcing enables users to modify prompts during streaming to interactively generate video in real-time.

attention costs compared to full attention over 21 frames.

The latency range after first rolling in Table 7 reflects this trade-off. While Deep Forcing exhibits a wider latency range (0.747s to 0.797s) compared to the baseline (0.770s to 0.776s), the average latencies are nearly identical, demonstrating that compression overhead is effectively balanced by reduced attention costs. In practice, throughput oscillates between compression phases (slightly slower) and generation phases (slightly faster) as the cache alternates between 21 and 16 frames. These fluctuations average to nearly identical performance as the baseline, demonstrating that our compression mechanism effectively amortizes its overhead, enabling long-horizon generation with minimal performance penalty.

## F. Interactive Prompting

Following LongLive (Yang et al., 2025b), we demonstrate that Deep Forcing remains compatible with interactive prompting, allowing users to modify prompts during generation to steer video content. When a prompt change occurs, we preserve the sink tokens and recompute the remaining KV cache with the new prompt, similar to LongLive's recache strategy.

As shown in Fig. 16, our method successfully handles diverse interactive scenarios without additional finetuning: (i) adding new subjects through prompt updates (e.g., a new character entering the scene), and (ii) camera transitions such as

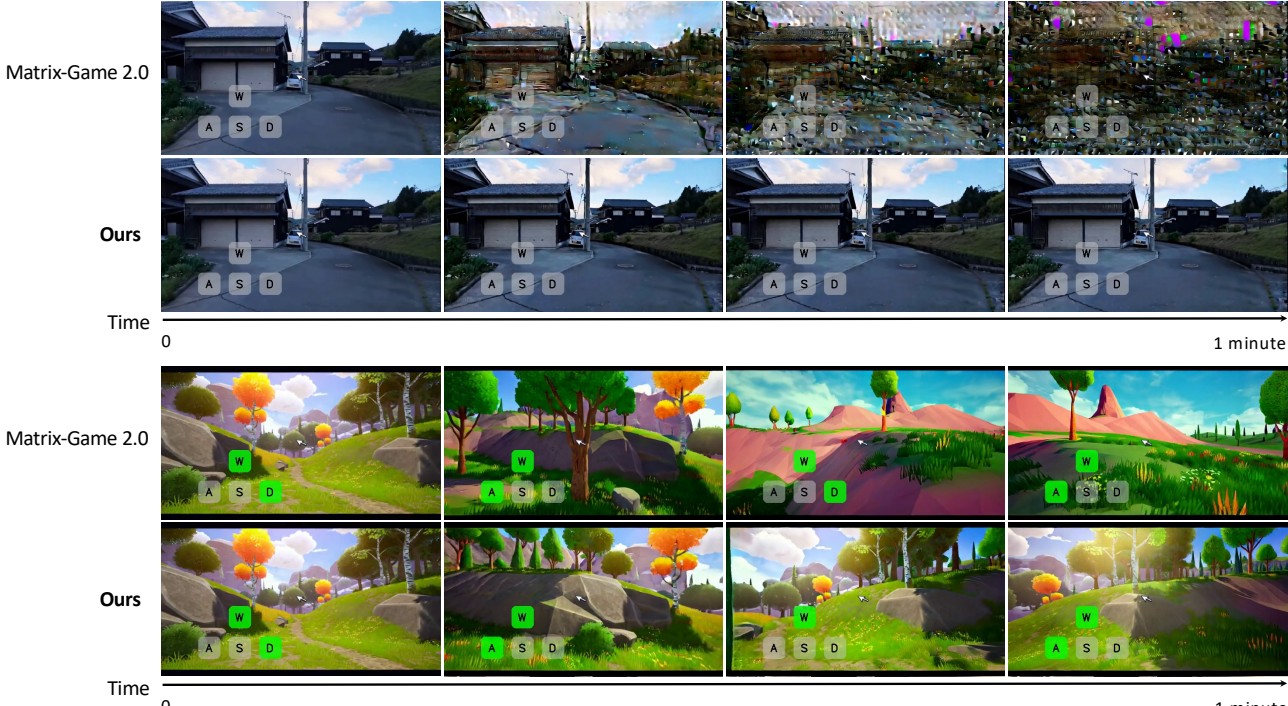

Figure 17. **Deep Forcing on World Model** Our method mitigates error accumulation & color drift in Matrix-Game 2.0 (He et al., 2025).

zoom-ins that shift to entirely different visual content (e.g., from a wide shot to a close-up of a photograph or chart). These results demonstrate that Deep Forcing remains robust in interactive prompting, even when the generated content diverges significantly from the sink frames.

## G. Deep Forcing on World Model

Self Forcing has recently been adopted in autoregressive diffusion world models (Ye et al., 2025; He et al., 2025; Tang et al., 2025). We apply our training-free method to Matrix-Game 2.0 (He et al., 2025), an open-source world model, and observe that it effectively minimizes error accumulation. As shown in Fig. 17, vanilla Matrix-Game 2.0 exhibits noticeable appearance drift over nearly a 1-minute rollout: the overall color/style gradually degrades, indicating error accumulation in long-horizon autoregressive generation.

In the upper pair, when no action condition is provided, vanilla Matrix-Game 2.0 (He et al., 2025) exhibits severe degradation over time. However, with our Deep Forcing framework, as shown in Fig. 17, degradation is substantially reduced even without input conditions. This demonstrates that Deep Forcing effectively maintains visual stability in world models during long-horizon generation.

In the lower pair, when action conditions are provided in real time, vanilla Matrix-Game 2.0 (He et al., 2025) exhibits degradation in the form of color drift and style shift. In contrast, applying Deep Forcing substantially reduces such visual fidelity degradation.

## H. Difference with FramePack

Recent work, FramePack (Zhang et al., 2026), also aims to improve long video generation with history compression, but it operates at a fundamentally different level and makes distinct design choices.

**FramePack: Frame-level input compression.** FramePack controls how history frames are represented before attention. It assigns frame-level importance ranks and compresses history into a bounded context via multi-rate patchification: older or less important frames are encoded with coarser patches, reducing their token count while maintaining bounded context length. This compression operates uniformly within each frame—it does not selectively preserve specific regions. Additionally,

FramePack optionally uses non-causal sampling strategies (e.g., bidirectional infilling) and requires finetuning to adapt the backbone to packed-history representations.

**Deep Forcing: Token-level KV cache compression.**   In contrast, our method targets *causal, streaming autoregressive* generation and operates *training-free* at inference time. Rather than altering input tokenization, we directly control *how the model retains memory* by operating on the KV cache. Deep Sink maintains persistent sink tokens with realigned temporal RoPE phases, stabilizing attention over long rollouts. Participative Compression performs *token-level* selection: it ranks individual KV caches using attention scores from recent queries and retains only the most relevant tokens, concentrating the cache budget on important regions (Fig. 13) within frames rather than preserving full-frame context uniformly.

**Summary.**   FramePack compresses history at the input level via frame-level ranking and uniform multi-rate patchification, often coupled with bidirectional infilling and finetuning. Deep Forcing operates at the KV cache level via token-level pruning.

## I. Design Justification of Deep Forcing

In this section, we provide additional justification for the two core components of Deep Forcing—Deep Sink (DS) and Participative Compression (PC). Deep Forcing is built upon two key findings: (1) video diffusion requires a substantially deeper attention sink than LLMs, and (2) Deep Sink alone is insufficient to fully mitigate error accumulation under extreme length extrapolation, motivating importance-aware pruning.

**Why Deep Sink?**   Motivated by StreamingLLM (Xiao et al., 2023), we analyzed the attention behavior of pre-trained Self Forcing (Huang et al., 2025) and observed that the model attends across a deep range of both initial and intermediate tokens (Fig. 4, Appendix T)—unlike LLMs, where only a small set of initial tokens serve as sinks. This motivates preserving a larger portion of early tokens as the sink. The design is further justified by the intuition that earlier frames lie within the training horizon and therefore carry relatively less accumulated error; maintaining a larger early sink keeps generation closer to the model's training-time distribution during long-horizon rollouts.

**Can a unified attention-score-based selection recover sink tokens?**   A natural question is whether the role of DS can be absorbed into PC by simply applying score-based selection over the entire cache, without explicitly preserving the earliest, least-error-accumulated tokens as a fixed sink. To test this, we conducted an ablation in which we removed the fixed sink and instead allocated the equivalent budget to additional Top-$C$ selection over the entire cache.

*Table 8.* **Ablation on unified selection without Deep Sink.** Removing the explicit sink and reallocating its budget to Top-$C$ selection degrades performance across all VBench-Long metrics on 30-second videos.

| Method | Dynamic Degree | Motion Smoothness | Overall Consistency | Imaging Quality | Aesthetic Quality | Subject Consistency | Background Consistency |
|---|---|---|---|---|---|---|---|
| Unified Selection (w/o DS) | 53.00 | 97.58 | 20.35 | 68.79 | 59.51 | 96.41 | 96.06 |
| Deep Forcing (Ours) | **57.56** | **98.27** | **20.54** | **69.31** | **60.68** | **97.34** | **96.48** |

As shown in Tab. 8, the unified-selection variant performs noticeably worse than our default design across all metrics. As Fig. 4 shows, attention weights on the earliest tokens are not substantially higher than those on intermediate tokens (unlike in LLMs, where few initial sinks dominate). Consequently, score-based selection without an explicit sink fails to reliably preserve the cleanest, least-error-accumulated tokens, often evicting them in favor of more recently attended intermediate ones. This empirically confirms that Deep Sink is necessary and cannot be replaced by importance-aware selection alone.

**Why Participative Compression?**   While Deep Sink effectively mitigates fidelity degradation within moderate extrapolation lengths, it alone cannot fully prevent quality degradation in minute-long generation. This phenomenon is well-documented in autoregressive long-context generation (Ghadia et al., 2025; Holtzman et al., 2019): beyond the training length, the growing KV cache accumulates increasingly irrelevant tokens, diluting attention across both relevant and irrelevant context and amplifying accumulated noise. Recent analysis of video diffusion further shows that attention concentrates on a small subset of semantically critical tokens, with the majority contributing minimally to generation quality (Yang et al., 2025c), suggesting that pruning low-attention tokens can reduce error accumulation with limited quality

impact. These observations motivate PC, which dynamically retains only contextually relevant tokens while pruning those that contribute to attention dilution and noise propagation.

**Why preserve recent tokens?** The recent tokens $(K_{\text{rct}}, V_{\text{rct}})$ excluded from compression serve two complementary roles: (a) as a *query proxy* for computing Top-$C$ importance scores—without recent queries, PC has no basis for ranking candidate tokens by their relevance to the current generation context; and (b) as an anchor for *local temporal coherence*, ensuring smooth transitions across consecutive frames. As shown in Tab. 5, combining clean past queries with the current denoising queries yields the most effective Top-$C$ selection.

## J. Robustness to Hyperparameter Choice

We selected sink size $S{=}10$, budget $N{=}16$, and recent $R{=}4$ as our default configuration because it yielded the best balanced performance across all VBench-Long dimensions. However, our method is not brittle to this specific choice; it exhibits a broad stable operating region. Tab. 6 (Appendix C) varies budget and recent with sink fixed, while Tab. 9 below additionally varies sink and recent with budget fixed.

*Table 9.* **Robustness across Deep Forcing configurations.** Varying sink ($S$), recent ($R$), and budget ($N$) yields comparable VBench-Long performance, indicating a broad stable operating region. Evaluated on 30-second videos.

| Config | Dynamic Degree ↑ | Motion Smoothness ↑ | Overall Consistency ↑ | Imaging Quality ↑ | Aesthetic Quality ↑ | Subject Consistency ↑ | Background Consistency ↑ |
|---|---|---|---|---|---|---|---|
| $S{=}5,\ R{=}9,\ N{=}16$ | 60.10 | 97.89 | 20.23 | 68.51 | 59.62 | 96.92 | 96.26 |
| $S{=}7,\ R{=}7,\ N{=}16$ | 57.59 | 97.88 | 20.36 | 68.54 | 59.94 | 96.23 | 96.41 |
| $S{=}8,\ R{=}4,\ N{=}16$ | 59.81 | 97.93 | 20.55 | 68.35 | 59.77 | 97.04 | 96.21 |
| $S{=}9,\ R{=}5,\ N{=}16$ | 58.86 | 97.96 | 20.51 | 69.61 | 60.56 | 96.95 | 96.50 |
| $S{=}10,\ R{=}4,\ N{=}16$ *(Default)* | 57.56 | 98.27 | 20.54 | 69.31 | 60.68 | 97.34 | 96.48 |
| $S{=}11,\ R{=}4,\ N{=}16$ | 56.86 | 98.02 | 20.78 | 68.62 | 60.32 | 97.29 | 96.34 |

Overall, the method works well under two simple conditions: (i) the sink covers a substantial portion of the context window—deeper sinks retain more low-error early tokens, reducing drift accumulation (consistent with Fig. 5)—and (ii) the recent window is kept relatively small to most effectively identify tokens relevant to ongoing generation.

## K. Transferability to Causal Forcing

Beyond Self Forcing (Huang et al., 2025) and Matrix-Game 2.0 (He et al., 2025) (Appendix G), we further verify that Deep Forcing transfers to Causal Forcing (Zhu et al., 2026), a recent autoregressive diffusion distillation method built on the same architecture. Tab. 10 shows results on 30-second videos with Causal Forcing as the base model. Across a wide range of configurations, applying Deep Forcing yields improvements over the original Causal Forcing across nearly all metrics at our default setting ($S{=}10$, $R{=}4$, $N{=}16$). This confirms that the design of Deep Forcing generalizes beyond the Self Forcing used in our main experiments, and is applicable to other autoregressive video diffusion model. Additional qualitative results are available on our project page.[1]

## L. Ablation on Per-Timestep Top-$C$ Selection

By default, Deep Forcing performs Top-$C$ selection only once at the first diffusion timestep ($t{=}T$) and reuses the selected tokens throughout the remaining denoising steps. To validate this design choice, we additionally evaluate a variant that performs Top-$C$ selection independently at every diffusion timestep. As shown in Tab. 11, the per-timestep variant achieves similar performance to our default single-timestep design. This result is consistent with Fig. 12, which shows that attention patterns remain highly consistent across timesteps—tokens deemed important at the initial timestep remain important throughout the denoising process. Importantly, per-timestep selection requires maintaining separate KV caches for each diffusion timestep, leading to additional VRAM consumption with no quality benefit. We therefore adopt single-timestep selection at $t{=}T$ as our default for efficiency.

---

[1] https://cvlab-kaist.github.io/DeepForcing/

*Table 10.* **Transferability of Deep Forcing to Causal Forcing.** Applying Deep Forcing to Causal Forcing yields consistent improvements across a wide range of configurations on 30-second videos. "Original" denotes the Causal Forcing baseline.

| Config | Dynamic Degree ↑ | Motion Smoothness ↑ | Overall Consistency ↑ | Imaging Quality ↑ | Aesthetic Quality ↑ | Subject Consistency ↑ | Background Consistency ↑ |
|---|---|---|---|---|---|---|---|
| Original (Causal Forcing) | 68.99 | 97.12 | 18.66 | 65.25 | 55.44 | 95.74 | 95.58 |
| $S$=10, $R$=4, $N$=15 | 81.06 | 96.95 | 19.95 | 69.96 | 60.22 | 96.42 | 95.73 |
| $S$=10, $R$=4, $N$=16 *(Default)* | 80.41 | 97.16 | 20.15 | 70.03 | 60.62 | 96.26 | 95.60 |
| $S$=10, $R$=5, $N$=16 | 79.95 | 97.27 | 20.31 | 70.32 | 60.36 | 96.17 | 95.70 |
| $S$=10, $R$=4, $N$=17 | 81.45 | 96.78 | 19.55 | 68.31 | 58.34 | 96.16 | 95.83 |
| $S$=10, $R$=5, $N$=17 | 81.84 | 97.04 | 19.23 | 68.37 | 59.31 | 96.28 | 95.36 |
| $S$=10, $R$=6, $N$=17 | 81.04 | 97.07 | 19.57 | 68.21 | 59.43 | 95.78 | 95.75 |
| $S$=10, $R$=4, $N$=18 | 80.02 | 96.55 | 19.70 | 67.56 | 59.88 | 95.73 | 95.48 |

*Table 11.* **Ablation on per-timestep Top-$C$ selection.** Performing Top-$C$ selection at every diffusion timestep yields nearly identical performance to our default single-timestep selection at $t$=$T$.

| Method | Dynamic Degree ↑ | Motion Smoothness ↑ | Overall Consistency ↑ | Imaging Quality ↑ | Aesthetic Quality ↑ | Subject Consistency ↑ | Background Consistency ↑ |
|---|---|---|---|---|---|---|---|
| Per-Timestep Selection | 59.16 | 98.18 | 20.52 | 69.35 | 60.01 | 97.28 | 96.51 |
| Deep Forcing (Ours, $t$=$T$ only) | 57.56 | 98.27 | 20.54 | 69.31 | 60.68 | 97.34 | 96.48 |

# M. Justification of Attention-Based Token Selection

**Why attention-based importance?** The goal of Participative Compression (PC) is not simply to shrink the KV cache, but to dynamically preserve historical tokens that remain actively relevant to the current generation. We therefore define the importance score as the attention from recent queries to candidate keys, so that retained tokens are those the model is currently relying on. This makes the selection criterion adaptive and model-aligned, unlike FIFO eviction. To validate this design choice, we compare against two alternative selection strategies: (1) *random Top-$C$ selection*, which causes artifacts including drift and incoherent transitions (Appendix D, Fig. 14), and (2) *key-similarity selection*, which ranks candidate tokens by the cosine similarity between recent keys and historical keys. As shown in Tab. 12, key-similarity selection underperforms our attention-based design, because feature resemblance alone does not capture current importance.

*Table 12.* **Ablation on token selection strategy.** Key-similarity selection (cosine similarity between recent and historical keys) underperforms our attention-based selection.

| Method | Dynamic Degree ↑ | Motion Smoothness ↑ | Overall Consistency ↑ | Imaging Quality ↑ | Aesthetic Quality ↑ | Subject Consistency ↑ | Background Consistency ↑ |
|---|---|---|---|---|---|---|---|
| Key-Similarity Selection | 62.15 | 98.10 | 20.47 | 68.78 | 60.25 | 97.04 | 96.31 |
| Deep Forcing (Ours) | 57.56 | **98.27** | **20.54** | **69.31** | **60.68** | **97.34** | **96.48** |

# N. Additional Quantitative Evaluation

To further validate the robustness of our evaluation, we report two additional quantitative analyses: Standard VBench results (without the temporal smoothing of VBench-Long) and HSV-based color consistency measurements on 30-second generated videos.

**Standard VBench results.** We report Standard VBench results (Tab. 13), which evaluate single-clip quality without the temporal smoothing of VBench-Long (Huang et al., 2024). As shown in Tab. 13, the overall trends are consistent with our VBench-Long evaluations (Tab. 1): Deep Forcing achieves the highest Dynamic Degree while maintaining Imaging Quality and Aesthetic Quality comparable to training-based methods such as Rolling Forcing (Liu et al., 2025) and LongLive (Yang et al., 2025b). Notably, unlike in VBench-Long, Deep Forcing outperforms CausVid (Yin et al., 2025) in both Subject Consistency and Background Consistency, which better aligns with the perceptual quality observed in our user study (Tab. 4).

**HSV color consistency.** To complement the metrics in VBench, we additionally measure HSV-based color consistency on 30-second generated videos. We convert each frame to HSV color space and compute a 3D histogram over the Hue, Saturation, and Value channels. We report two metrics: (1) *Consecutive Correlation*, the average histogram correlation

*Table 13.* **Standard VBench results on 30-second videos.** Standard VBench evaluates single-clip quality without the temporal smoothing applied in VBench-Long. Deep Forcing achieves the highest Dynamic Degree while maintaining quality metrics comparable to training-based baselines.

| Method | Dynamic Degree ↑ | Motion Smoothness ↑ | Overall Consistency ↑ | Imaging Quality ↑ | Aesthetic Quality ↑ | Subject Consistency ↑ | Background Consistency ↑ |
|---|---|---|---|---|---|---|---|
| Rolling Forcing | 32.81 | 98.73 | 21.79 | 70.88 | 62.06 | 93.52 | 94.22 |
| LongLive | 52.34 | 98.74 | 20.93 | 69.46 | 63.41 | 92.09 | 94.03 |
| CausVid | 51.56 | 98.05 | 20.44 | 66.66 | 61.37 | 87.28 | 90.13 |
| Self Forcing | 42.96 | 98.61 | 21.37 | 68.95 | 61.00 | 87.79 | 91.16 |
| Deep Forcing (Ours) | 60.71 | 98.25 | 21.59 | 69.63 | 62.55 | 91.51 | 93.21 |

between adjacent frames, measuring short-term color stability; and (2) *First-Last Correlation*, the histogram correlation between the first and last frames, capturing long-range color drift over the full 30-second duration. As shown in Tab. 14, Deep Forcing substantially improves HSV color consistency over Self Forcing and CausVid, while remaining competitive with training-based methods such as Rolling Forcing and LongLive. In particular, the large gain in First-Last Correlation over Self Forcing demonstrates that our training-free approach effectively mitigates the long-range color drift commonly observed in the underlying Self Forcing base model, preserving color coherence throughout long-horizon generation.

*Table 14.* **HSV color consistency on 30-second videos.** *Consecutive Correlation* measures short-term color stability between adjacent frames, while *First-Last Correlation* captures long-range color drift between the first and last frames. Deep Forcing substantially mitigates the long-range color drift observed in Self Forcing.

| Method | Consecutive Correlation ↑ | First-Last Correlation ↑ |
|---|---|---|
| CausVid | 96.68 | 20.64 |
| Self Forcing | 94.12 | 14.93 |
| LongLive | 98.02 | 42.92 |
| Rolling Forcing | 97.25 | 54.27 |
| Deep Forcing (Ours) | 97.55 | 48.75 |

Together, the Standard VBench and HSV color consistency results are consistent with our main VBench-Long evaluation, further validating the robustness of our analysis across complementary evaluation protocols.

## O. VLM-based Verification of Motion Quality

A natural concern with our higher Dynamic Degree scores is whether the additional motion reflects genuinely richer dynamics or hallucinated motion artifacts. To directly investigate this, we use Gemini 3.1-Pro (Comanici et al., 2025) to classify each of the 128 benchmark videos as exhibiting either *genuinely richer dynamics* or *hallucinated motion or artifacts*. Importantly, we also classify videos generated by Self Forcing (Huang et al., 2025) under the same protocol as a baseline reference, allowing us to measure the relative change in the artifact rate introduced by our method rather than relying on absolute classifier confidence.

*Table 15.* **VLM-based motion classification.** Gemini 3.1-Pro classifies each generated video as exhibiting genuinely richer motion or hallucinated motion/artifacts. Deep Forcing largely preserves the base model's quality distribution.

| Method | Genuinely Richer Motion (%) ↑ | Hallucinated Motion (%) ↓ |
|---|---|---|
| Self Forcing | 87.4 | 12.6 |
| Deep Forcing (Ours) | 83.6 | 16.4 |

As shown in Tab. 15, Self Forcing yields an 87.4%:12.6% (richer motion : hallucinated motion) ratio, while Deep Forcing yields 83.6%:16.4%. While Deep Forcing exhibits a slight increase in hallucinated motion rate, the overall quality distribution remains within a similar regime as the base model, indicating that our method does not introduce substantial additional artifacts beyond those already present in Self Forcing. Furthermore, among Deep Forcing videos classified as exhibiting richer motion, the average Dynamic Degree is 58.17, compared to 55.05 for those classified as hallucinated—confirming that the higher Dynamic Degree score is not primarily driven by artifacts. The prompt we use for evaluating is as following:

```
# Role: AI Video Quality Auditor

# Context:
You are evaluating the motion characteristics of AI-generated
videos. In this domain, minor visual inconsistencies are inherent
properties. Your evaluation should prioritize identifying whether
the motion is a meaningful manifestation of the prompt or a
byproduct of generative instability.

# Task:
Analyze the video to determine the *primary driver* of the
observed motion. Distinguish whether the motion represents
"Genuinely Richer Dynamics" or if it is primarily
"Hallucinated Motion."

# Step 1: Motion Source Classification
Classify the video based on the following structural criteria:

[Category A] Genuinely Richer Dynamics
- The motion originates from identifiable actions or camera work
  that maintains alignment with the prompt's semantic intent.
- Criteria: The subject maintains structural coherence and the
  motion follows a discernible, purposeful trajectory throughout
  the video.

[Category B] Hallucinated Motion / Artifact
- The motion fundamentally lacks semantic alignment with the
  prompt. High pixel-level change is predominantly driven by
  structural disintegration, severe anatomical distortion (e.g.,
  limbs splitting into multiple copies, faces collapsing into
  unrecognizable forms), or persistent full-frame corruption
  that renders the scene uninterpretable.
- Criteria: The movement lacks any logical trajectory, or the
  subject's structure is severely compromised such that a human
  observer cannot track any purposeful action.

# Input Information:
- Video Generation Prompt: [Insert Prompt Here]

# Output Format:
1. Observation: (Briefly describe the motion and its trajectory.
   State whether the motion follows a recognizable path.)
2. Classification: (Strictly output either "Genuinely Richer
   Dynamics" or "Hallucinated Motion")
```

## P. Detailed Comparison with LLM KV Cache Methods

While our method is inspired by KV cache management techniques developed for large language models, Deep Forcing introduces a series of design choices specifically tailored to autoregressive video diffusion. In this section, we detail the structural differences between our method and each closely related LLM method.

**Deep Sink vs. StreamingLLM (Xiao et al., 2023).** StreamingLLM shows that retaining only 4 initial tokens is sufficient for LLMs, and discards all intermediate tokens. Prior autoregressive video methods such as LongLive (Yang et al., 2025b) and Rolling Forcing (Liu et al., 2025) adopted this LLM convention without verifying it for video. Our attention analysis on autoregressive video diffusion (Fig. 4) reveals a fundamentally different pattern: Self Forcing distributes substantial attention not only over the earliest frames but also over intermediate frames across the sequence. This necessitates a deeper sink covering 40–60% of the context window, in contrast to the few-token sink convention in LLMs.

**Participative Compression vs. H2O (Zhang et al., 2023).** H2O retains tokens with the highest *cumulative* attention scores aggregated across the entire generation history. In video generation, however, visual relevance evolves continuously: tokens important for earlier frames may no longer matter for the current frame. PC instead scores tokens using only the queries of the most recent $R$ frames, directly capturing what the model is currently relying on rather than what it has cumulatively relied on in the past.

**Participative Compression vs. SnapKV (Li et al., 2024).** SnapKV performs *one-shot* prompt compression during the prefill stage, fixing the retained tokens based on attention patterns within the input prompt before generation begins. All tokens generated during decoding are subsequently retained without further compression, so the cache grows unboundedly with output length. PC, in contrast, operates *during* decoding: it iteratively compresses the rolling KV cache whenever the sliding window reaches its limit, maintaining a bounded cache size while continuously adapting to the evolving context of long video generation.

**Participative Compression vs. MorphKV (Ghadia et al., 2025).** MorphKV dynamically selects tokens based on already-generated clean tokens. PC differs in three important ways. First, PC additionally leverages the *currently denoising* query at $t{=}T$ as a selection signal—noisy but informative (Tab. 5)—enabling selection that anticipates what the model is about to generate rather than only what it has already generated. Second, PC compresses only at the first diffusion timestep when the window overflows, rather than at every chunk generation, exploiting the cross-timestep stability of attention patterns we identify in Fig. 12. Third, after compression, PC applies temporal RoPE unification to the selected tokens to prevent temporal discontinuities—a step that does not arise in the LLM setting.

**Summary.** The novelty of Deep Forcing lies in: (a) showing that autoregressive video diffusion requires fundamentally deeper sinks than LLMs, co-designed with importance-aware compression in a three-partition cache structure (sink, candidate, recent); and (b) addressing video-specific challenges absent in the LLM setting, including 3D temporal RoPE adjustment, denoising-query-based selection, and diffusion-timestep-aware compression. These design choices are not direct adaptations of existing LLM techniques but rather a reformulation tailored to the unique demands of long-horizon autoregressive video generation.

## Q. Inter-Rater Agreement in User Study

To assess the reliability of our user study (Tab. 4), we compute Fleiss' $\kappa$ for all pairwise comparisons. We interpret the resulting values using the widely adopted Landis-Koch convention (Landis & Koch, 1977): 0–0.20 slight, 0.21–0.40 fair, 0.41–0.60 moderate, 0.61–0.80 substantial, and 0.81–1.00 almost perfect agreement.

*Table 16.* **Inter-rater agreement (Fleiss' $\kappa$) in our user study.** Agreement levels are interpreted using the Landis-Koch convention.

| vs. Baseline | Fleiss' $\kappa$ | Agreement Level |
|---|---|---|
| CausVid | $\approx 1.00$ | Almost perfect |
| Self Forcing | 0.83 | Almost perfect |
| LongLive | 0.50 | Moderate |
| Rolling Forcing | 0.38 | Fair |

As shown in Tab. 16, agreement is almost perfect against weaker baselines (CausVid, Self Forcing) and decreases against stronger training-based baselines (LongLive, Rolling Forcing). This pattern is expected: stronger baselines produce finer perceptual differences relative to Deep Forcing, which naturally leads to more divided individual judgments. Importantly, however, Deep Forcing was consistently preferred across all four evaluation criteria even in these closer comparisons (Tab. 4), demonstrating that the preference signal remained directional.

## R. More Qualitative Results

Additional qualitative results of our method are presented in Figure 18, and Figure 19. These examples clearly show that our **training-free** Deep Sink and Participative Compression framework produces results on par with training-based methods.

## S. User Study Protocol

To perform human evaluation, we conducted a user study based on the Two-Alternative Forced Choice (2AFC) protocol (Blattmann et al., 2023; Chefer et al., 2025). For each question, participants were presented with two videos generated from the same prompt and instructed to choose which video they preferred according to four evaluation criteria: (1) Color Consistency - Which video maintains more consistent color and exposure throughout, without sudden shifts in brightness, saturation, or color tone? (2) Dynamic Motion - Which video exhibits more dynamic and varied motion, including both subject movement and camera movement? (3) Subject Consistency - Which video maintains better visual consistency of the main subject throughout its duration? (Consider comparing the beginning and end of each video.) (4) Overall Quality - Overall, which video appears more realistic, natural, and of higher quality?

Each participant evaluated 16 video pairs comparing Deep Forcing against each of the four baselines (CausVid (Yin et al., 2025), Self Forcing (Huang et al., 2025), LongLive (Yang et al., 2025b), and Rolling Forcing (Liu et al., 2025)). For each baseline, participants were shown 4 pairwise comparisons using different prompts, with all 16 prompts being non-overlapping within each participant's session. These prompts were randomly sampled from a pool of 20 total prompts. With 24 total participants, this yielded 384 total video comparisons (24 participants × 16 pairs), the results of which are shown in Table 4. The presentation order of videos was randomized, and participants were not informed which model generated each video. This design ensured balanced evaluation across all baseline models while avoiding prompt repetition within individual sessions. The user interface is shown in Figure 21.

## T. Additional Attention Visualization

We provide additional attention head visualizations (Fig.20) beyond those shown in Fig.4 from Section 4.1. This deep attention pattern with substantial weight on both initial and intermediate tokens, emerges consistently and pervasively across layers and heads, rather than being only one or two specific heads, supporting the hypothesis that deep sinks are fundamental to Self Forcing (Huang et al., 2025).

## U. Limitations and Future Works

While our training-free approach substantially improves long-horizon stability, several limitations remain. Operating at inference time on a frozen backbone, our method is constrained by the pretrained model's capacity and biases. Additionally, our approach lacks explicit long-term memory, potentially causing gradual drift in extremely long sequences with repeated occlusions. Future work could integrate hierarchical memory modules and extend to broader video generation settings.

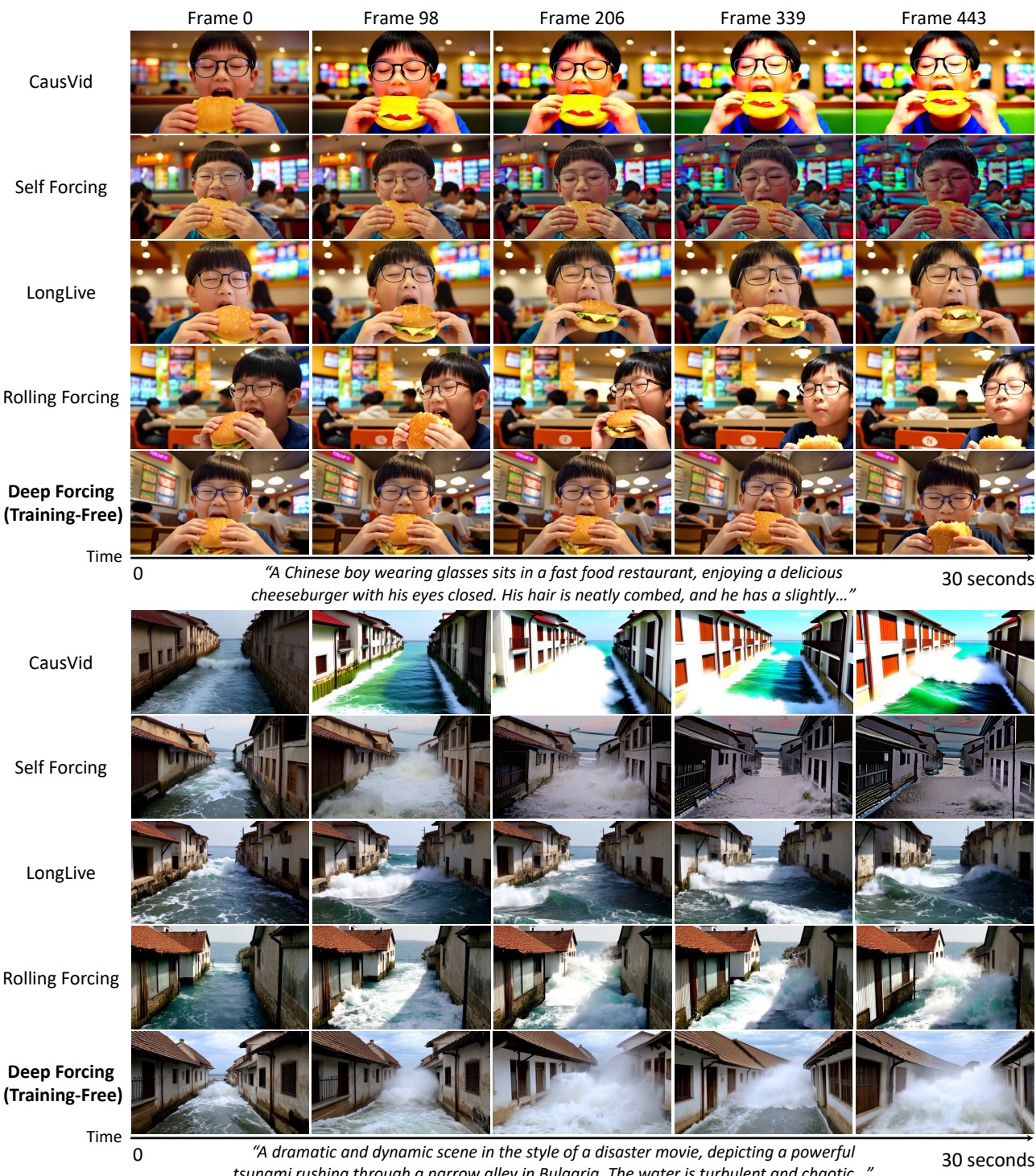

*Figure 18.* **Qualitative results on 30-second videos.** Frame-by-frame comparison across different methods for two representative prompts. Deep Forcing (training-free) achieves temporal consistency and visual quality comparable to training-based baselines (CausVid (Yin et al., 2025), Self Forcing (Huang et al., 2025), LongLive (Yang et al., 2025b), Rolling Forcing (Liu et al., 2025)) while generating more dynamic content with greater subject consistency.

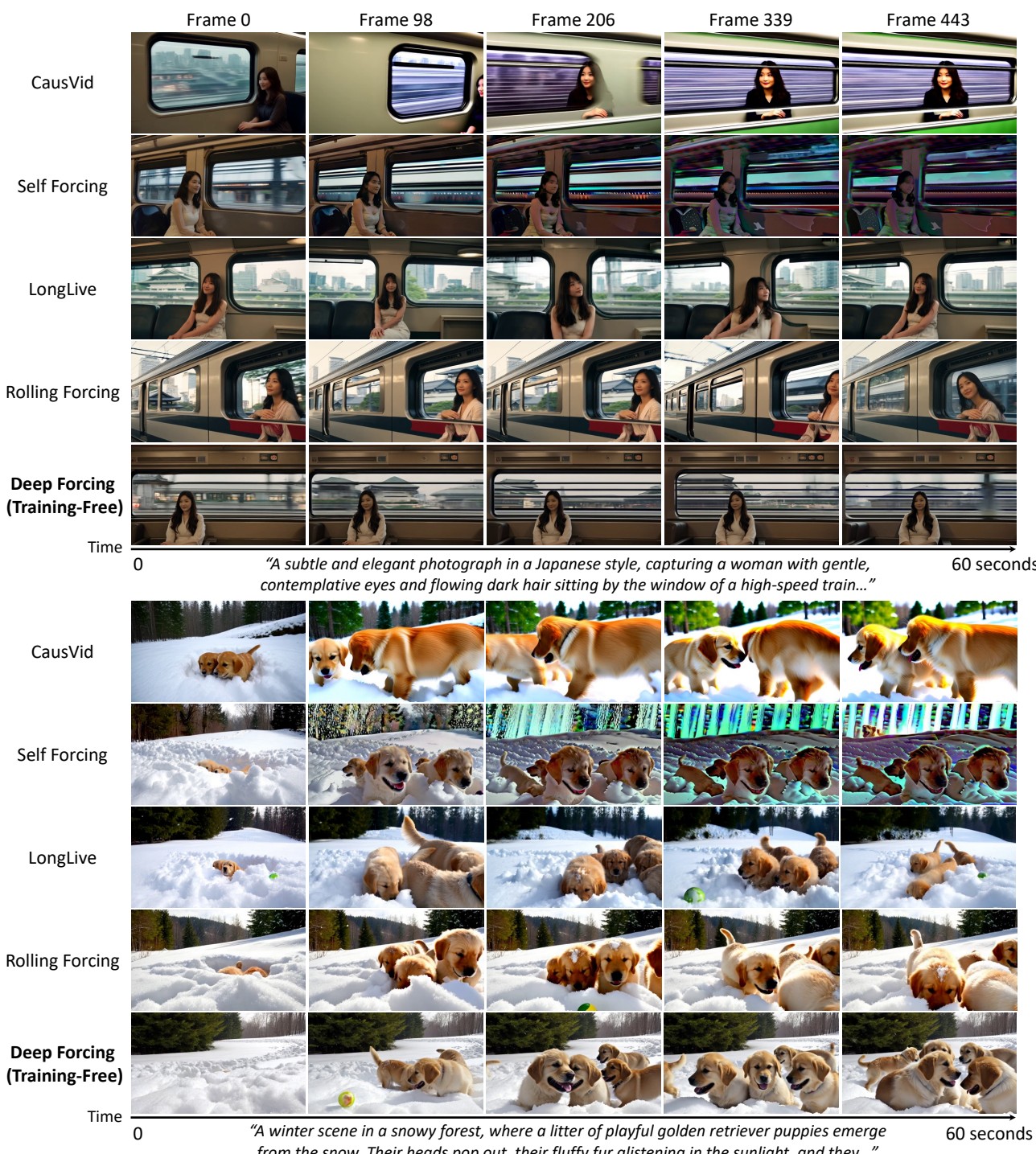

*Figure 19.* **Qualitative results on 60-second videos.** Frame-by-frame comparison across different methods for two representative prompts. Deep Forcing (training-free) achieves temporal consistency and visual quality comparable to training-based baselines (CausVid (Yin et al., 2025), Self Forcing (Huang et al., 2025), LongLive (Yang et al., 2025b), Rolling Forcing (Liu et al., 2025)) while generating more dynamic content with greater subject consistency.

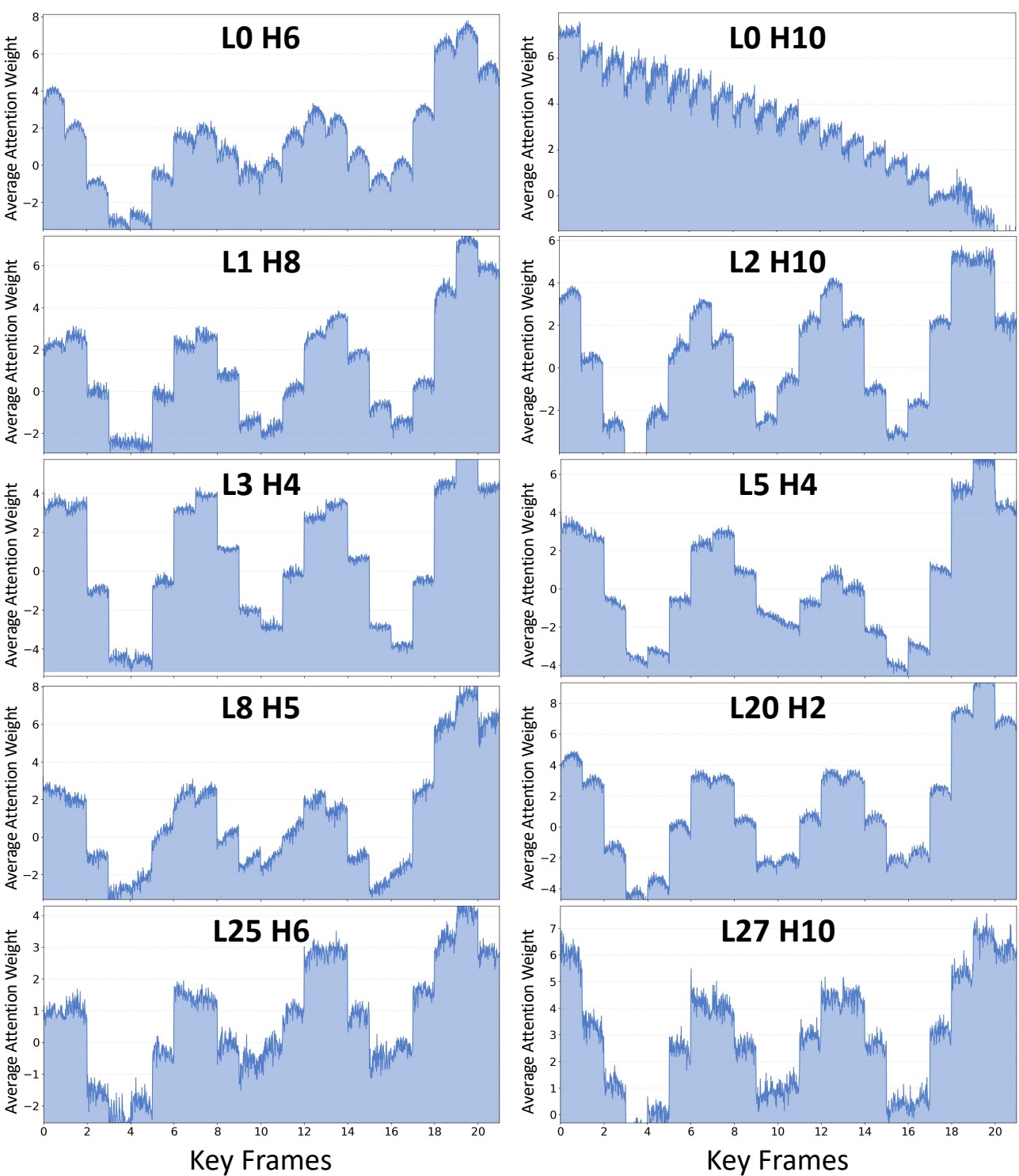

*Figure 20.* **Attention weight distribution across earlier frames.** Query-averaged attention showing how the last chunk (frames 19-21) attends to earlier KV cache entries (frames 0-18). Each frame consists of 1,560 tokens (spatially arranged latent patches). We visualize multiple attention heads from different layers, demonstrating that substantial attention to intermediate tokens is consistent across layers and heads.

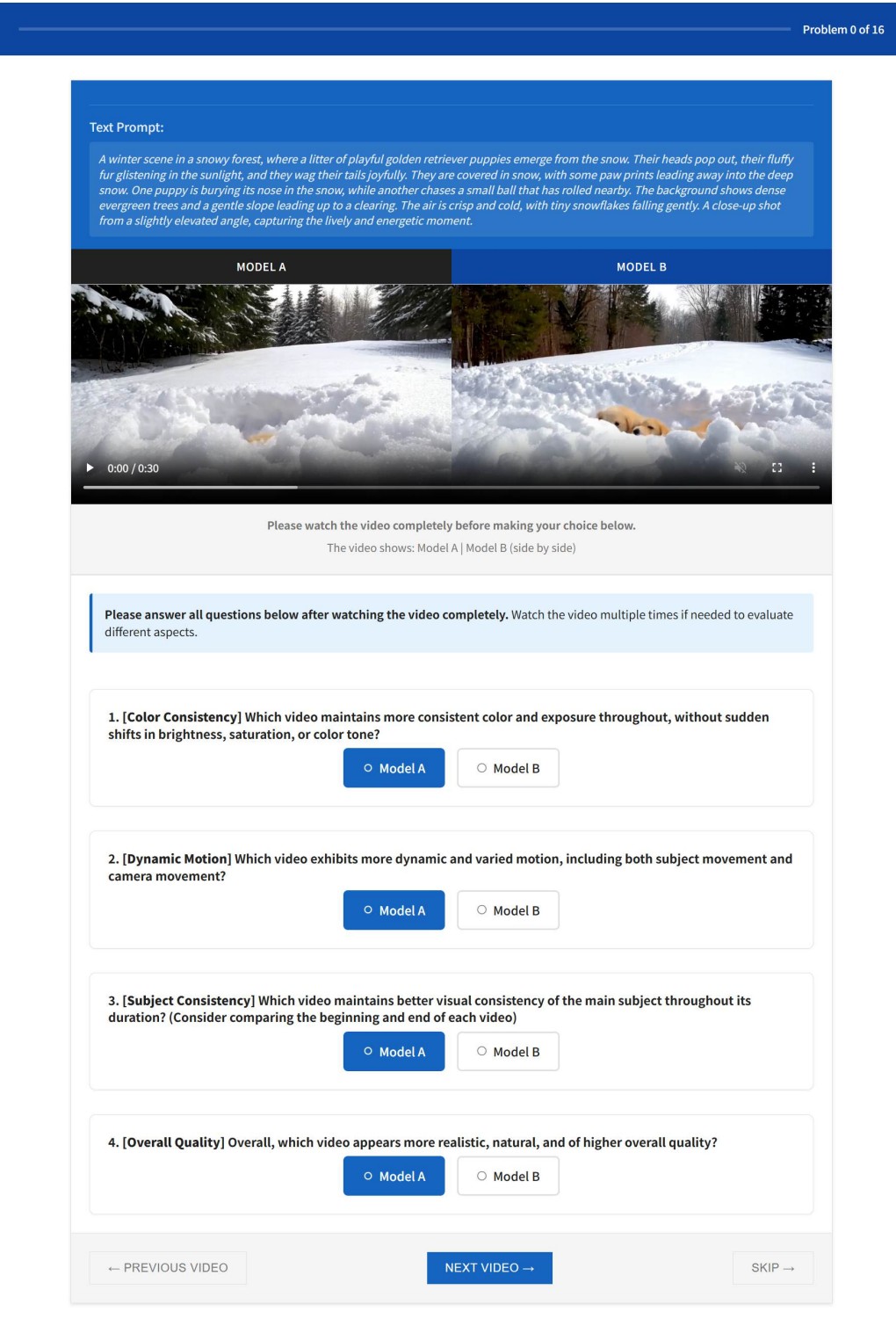

*Figure 21.* **Example of the user interface for the user study.**

