# OpenReview forum: "Deep Forcing: Training-Free Long Video Generation with Deep Sink and Participative Compression"
_ICML.cc/2026/Conference — ICML 2026 regular_

### Official Review · Reviewer_HoYe · 2026-02-16

**Soundness:** 3
**Presentation:** 3
**Significance:** 3
**Originality:** 3
**Overall Recommendation:** 4
**Confidence:** 5

**Summary:**

This paper proposes Deep Forcing, a training-free extension to autoregressive video diffusion that stabilizes long-horizon video generation via Deep Sink (persistent early-context tokens with temporal RoPE realignment) and Participative Compression (attention-based KV-cache pruning). Extensive experiments on benchmarks, user studies, and VLM-based evaluations demonstrate the effectiveness of Deep Forcing.

**Compliance With Llm Reviewing Policy:**

Affirmed.

**Final Justification:**

The authors have addressed my concerns, and I will keep my positive score.

**Key Questions For Authors:**

Please refer to W.1-W.3 for the specific questions.

**Limitations:**

yes

**Strengths And Weaknesses:**

**Strengths**

1. The motivation of Deep Forcing is very clear, and its claim (e.g., deep sink) is supported by interpretability analyses.

2. The proposed Deep Sink and Participative Compression modules are well-motivated.

3. The comparison and ablation studies are thorough, and the method also demonstrates broader applicability.

**Weaknesses**

1. Deep Forcing is mainly designed for Self-Forcing-like autoregressive (AR) video diffusion models. Can Deep Forcing also be applied to bidirectional video generation models that support long video generation, such as LongCat-Video.

2. The parameter scales of the baselines that Deep Forcing builds upon are not clearly specified in the paper. It is unclear how much gain Deep Forcing would provide on larger-scale models, such as SkyReels-V2-14B and MAGI-1-24B.

3. The paper lacks a dedicated failure-case analysis. It would be helpful to include representative failure examples and an analysis of when Deep Sink or PC fails.

---

> ### Author Rebuttal · Authors · 2026-03-31
>
> We sincerely thank the reviewer for the thorough evaluation and for recognizing the clarity of our motivation and the broader applicability of our method. We address each point below.
>
> **W1.** Deep Forcing is designed for Self-Forcing-style autoregressive video diffusion models, where frames are generated sequentially and past context is maintained through a rolling KV cache. In this setting, long-horizon degradation is tied to the accumulation of drift in cached historical tokens, which motivates both Deep Sink and Participative Compression.
> LongCat-Video follows a different inference paradigm. It formulates generation as video continuation using noise-free condition frames and noisy target frames, with block causal attention ensuring condition tokens are not influenced by noisy tokens. Its KV cache stores the condition tokens' features to reuse across diffusion sampling steps — a fixed conditioning context, fundamentally different from Self Forcing-style autoregressive generation.
> For this reason, Deep Forcing is not directly applicable: Deep Sink relies on preserving early clean tokens as persistent anchors within a rolling cache, and Participative Compression assumes a growing historical cache that must be selectively pruned under a budget. These assumptions do not hold in LongCat-Video's conditioning-based architecture. That said, the underlying intuition — allocating context budget wisely for long-range conditioning — may inspire analogous designs for such models, which we consider an interesting future direction.
>
> **W2.** We agree that clarifying the parameter scales is important. However, SkyReels-V2-14B and MAGI-1-24B employ substantially different inference paradigms that make Deep Forcing not directly applicable.
>
> Both models adopt Diffusion Forcing-style inference where only a few clean caches or latents are involved in the sliding window and the rest of frames(more than half of the window) perform joint denoising.
>
> This breaks Deep Sink's core assumption that the earliest 40~60% KV cache remains in the window. Each model has additional structural differences: SkyReels-V2 performs sliding-window diffusion-forcing generation, where overlapping frames from the previous segment condition the joint denoising of the current window under frame-wise asynchronous noise schedules; MAGI-1 performs pipelined chunk-wise denoising where up to 4 chunks at different denoising stages are processed concurrently via block-causal attention. Neither architecture uses the rolling KV cache that Deep Forcing targets.
>
> Nevertheless, within the clean-context autoregressive paradigm, we have demonstrated that Deep Forcing generalizes across Self Forcing, Causal Forcing [1], and the Self Forcing-based world model Matrix-Game 2.0, which leads us to believe these principles could extend to larger-scale models within the same paradigm. (Causal Forcing results in anonymous Link: https://deeprebuttal.github.io/deeprebutaal/ .)
>
> [1]: Zhu et al. "Causal Forcing: Autoregressive Diffusion Distillation Done Right for High-Quality Real-Time Interactive Video Generation” arxiv26
>
> **W3.** As with any video generation method, Deep Forcing is not without limitations. We have observed two primary failure modes:
>
> (1) **Multi-minute generation.** While Deep Forcing substantially improves stability for minute-long videos, Deep Sink and Participative Compression alone are not sufficient to maintain robust quality in multi-minute generation (e.g., 2–4 minutes).
>
> (2) **Base model limitations.** As a training-free method, Deep Forcing inherits the base model's limitations for certain prompt types. For instance, prompts requiring 3D-aware or physically-aware understanding, or those involving complex multi-subject interactions, remain challenging regardless of our KV cache management.
>
> Addressing these limitations — particularly extending robust generation to multi-minute horizons — is a promising direction for future work.
>
> We are grateful for the reviewer's constructive feedback and will incorporate all suggested improvements.

---

> > ### Author Rebuttal · Reviewer_HoYe · 2026-03-31
> >
> > Thank you for the rebuttal. The authors have addressed my concerns, and I will keep my positive score.

---

> > > ### Author Response · Authors · 2026-04-06
> > >
> > > We sincerely thank the reviewer for the thoughtful evaluation and for confirming that the concerns have been fully resolved. We would greatly appreciate it if the reviewer could consider reflecting this in the final rating.

---

### Official Review · Reviewer_bodP · 2026-03-07

**Soundness:** 3
**Presentation:** 2
**Significance:** 2
**Originality:** 2
**Overall Recommendation:** 3
**Confidence:** 3

**Summary:**

This paper proposes Deep Forcing, a training-free method for improving long-horizon autoregressive video generation built on top of pre-trained Self Forcing models. The method introduces two mechanisms: (1) Deep Sink, which retains approximately half of the sliding context window as persistent "sink" tokens (the earliest generated frames) and realigns their temporal RoPE positions to the current timeline, and (2) Participative Compression, an attention-score-based KV cache pruning strategy that selectively retains only the most important intermediate tokens while evicting redundant ones. The authors demonstrate on VBench-Long benchmarks (30s and 60s videos) that their training-free approach achieves performance comparable to training-based methods like Rolling Forcing and LongLive, with particular improvements in dynamic degree and visual fidelity, while maintaining the same inference throughput (~15.75 FPS).

**Compliance With Llm Reviewing Policy:**

Affirmed.

**Final Justification:**

I appreciate the authors’ detailed rebuttal and the additional analyses. In particular, the added discussion on why sink and recent tokens are explicitly preserved, as well as the additional ablations and robustness comments, improve the paper compared with the original submission. However, I would like to maintain my rate:

1. I still view the contribution as more of a well-executed engineering integration than a sufficiently strong methodological advance.
The paper combines and adapts several existing ideas around attention sinks, KV-cache selection/pruning, and cache management to the autoregressive video diffusion setting. While this adaptation appears practically useful and empirically effective, I am still not fully convinced that the resulting method constitutes enough conceptual novelty on top of prior sink- and attention-based cache methods.

2. My  concern about comparison fairness remains unresolved. The rebuttal explains why the authors did not include stronger matched baselines, and I understand the practical cost argument. However, this does not fully answer the more central question: whether similarly adapted or retrained training-based baselines, especially ones given comparable sink depth or comparable cache-management modifications, could recover some or much of the reported gains.

3. I also still feel that an important potential trade-off is under-discussed. From a modeling perspective, stronger sink preservation may help stabilize long-horizon generation, but it could also bias the model toward anchoring too strongly to earlier context, potentially reducing motion diversity or making videos feel comparatively more static. The paper reports strong dynamic-degree numbers, so I do not want to overstate this point, but I still think the work would benefit from a more direct discussion of whether improved long-term stability through sink-based retention comes with any trade-off in motion expressiveness, especially perceptually. At present, I do not think this issue is sufficiently analyzed or discussed. The demos shown by the supplementary are basically still without large motions.

I agree that the method is effective and that the additional evidence makes the paper stronger than before. However, I still do not feel confident that the work reaches the level of methodological contribution and empirical support that would justify a positive borderline recommendation. Overall, I appreciate the authors’ effort in the rebuttal and I think the paper has practical value, but my final judgment remains the same.

**Key Questions For Authors:**

1. The large jump in dynamic degree (36.62 -> 57.56) is presented as a strength, but could this partially reflect uncontrolled or hallucinated motion rather than genuinely richer dynamics? Have you analyzed failure cases where high dynamic degree corresponds to visual artifacts?

2. How would the training-based baselines perform if they were retrained with a 10-frame deep sink matching your configuration? The current comparison pits your optimally-configured inference-time method against baselines trained with a suboptimal (for inference-free use) 3-frame sink.

3. Your method shows equal or slightly lower subject consistency (97.34) and background consistency (96.48) compared to Self Forcing (97.34, 96.47) at 30s, and lower than training-based methods. The user study claims superiority in subject consistency. How do you reconcile this discrepancy? Could the user study preference be driven by other visual qualities rather than true subject consistency?

**Limitations:**

The authors briefly discuss limitations in Appendix , mentioning constraints from the frozen backbone and lack of explicit long-term memory. However, several important limitations are inadequately addressed: (1) no discussion of failure cases or when the method degrades, (2) no analysis of prompt types where the method underperforms,  (3) the potential for the "dynamic degree" improvement to mask motion artifacts is not acknowledged.

**Strengths And Weaknesses:**

**Strengths**
- S1: The paper addresses a practically important problem: error accumulation in autoregressive video generation during long-horizon rollouts. The training-free nature of the approach makes it immediately applicable to any Self Forcing-based model without retraining, which is valuable for practitioners.

- S2: The attention analysis in Section 4.1 and Figure 4 provides useful empirical motivation for the Deep Sink design. The observation that video diffusion models attend to intermediate frames (not just initial ones, as in LLMs) is an interesting finding specific to the video domain. The ablation on sink depth (Figure 5) is well-executed and informative.

**Weaknesses**

- W1: The core technical contributions — attention sinks and attention-score-based KV cache pruning — are well-established techniques from the LLM literature (StreamingLLM, H2O, SnapKV, MorphKV). The paper's novelty is primarily in applying these known techniques to the video diffusion setting with RoPE temporal adjustment. The "Deep Sink" is essentially a larger attention sink window with RoPE realignment, and "Participative Compression" is essentially SnapKV-style token selection. While the application is reasonable, the intellectual novelty is limited. The claim of being "the first training-free KV-cache compression method for autoregressive video diffusion" is a narrow novelty claim that relies on the recency of the video diffusion field rather than methodological depth.

- W2: The comparison is not entirely fair. The training-based baselines (Rolling Forcing, LongLive) were trained with only 3-frame attention sinks. When applied in a training-free setting (Appendix A, Figure 9), these baselines naturally underperform because their architectures were designed and trained with different sink configurations. A fairer comparison would be against baselines also retrained with deeper sinks, or at minimum, the paper should more clearly acknowledge this asymmetry. Additionally, subject consistency and background consistency scores are notably lower than or equal to Self Forcing for the proposed method, yet this is downplayed.


- W3: The user study design has potential issues. Only 24 participants with 4 pairwise comparisons per baseline per participant yields relatively few total comparisons per baseline (24 x 4 = 96). No inter-rater agreement statistics (e.g., Fleiss' kappa) are reported. The extremely high preference rates (e.g., 100% over CausVid for overall quality) suggest the study may not adequately control for trivial quality differences versus the specific contributions of the method.

- W4: The dynamic degree metric shows very large improvements (36.62 -> 57.56) but it is unclear whether higher dynamic degree is always desirable. The paper treats it as uniformly positive, but excessively dynamic videos may also indicate instability or hallucinated motion. No analysis distinguishes between "more dynamic because of richer content" versus "more dynamic because of artifacts or uncontrolled motion."

---

> ### Author Rebuttal · Authors · 2026-03-31
>
> **Q1, W4.** We thank the reviewer for raising this important point. Disentangling genuine dynamics from artifact-induced motion is a well-known open challenge in video generation evaluation, and we acknowledge that our method is not exempt from this concern.
>
> **Metric-level evidence.** Despite the substantial increase in dynamic degree (36.62 → 57.56), our motion smoothness (98.27), overall consistency (20.54), subject consistency (97.34), and background consistency (96.48) all remain comparable to Self Forcing and training-based baselines. If hallucinated motion were dominant, we would expect degradation across these correlated metrics.
>
> **VLM-based verification.** To directly analyze this concern, we used Gemini 3.1-Pro to classify each of our 128 benchmark videos as exhibiting either genuine richer dynamics or hallucinated motion/artifacts. (Due to the 5000-character limit we have ommited the prompt used. We will provide this in the revised paper.) Crucially, we also classified videos generated by Self Forcing under the same protocol as a baseline reference. Self Forcing yielded an 87.4% : 12.6% (Genuinely Richer Motion : Hallucinated Motion) ratio, while Deep Forcing yielded 83.6% : 16.4% — largely preserving the base models quality distribution rather than introducing substantial additional artifacts. Among videos classified as richer motion, the average dynamic degree was 58.17, compared to 55.05 for hallucinated, confirming that the higher dynamic degree is not exclusively driven by artifacts. While VLM-based evaluation is imperfect and a standardized benchmark for this distinction has yet to be established in the field, this analysis provides supplementary evidence that our improvement is not primarily artifact-driven.
>
> **Perceptual consistency.** Our user study (Table 4) further supports this — participants preferred Deep Forcing not only in dynamic motion but also in subject consistency and overall quality across all baselines, suggesting the increased dynamics do not come at the cost of other perceptual qualities.
>
> Qualitative results on our anonymous project page in supplementary material further corroborate these findings.
>
> **Q2, W2.** We acknowledge that our presentation in Appendix A may not have clearly conveyed our intent. The comparison was designed as a training-free ablation to demonstrate the importance of both Deep Sink depth and RoPE adjustment — not as a direct comparison against the trained versions of LongLive and Rolling Forcing. We named these configurations "LongLive-style" and "Rolling Forcing-style" simply to describe their sink settings (3-frame sink without RoPE, and 3-frame sink with RoPE, respectively), not to imply we were evaluating the trained models themselves. All experiments in Appendix A were conducted on the same pretrained Self Forcing model to isolate the effect of sink depth and RoPE adjustment. We will revise the naming and presentation to make this intention clearer. Our paper's central contribution is that training-free Deep Sink can achieve comparable performance to training-based methods, eliminating the need for costly retraining.
>
> **Q3.** We first clarify that Deep Forcing's subject consistency (97.34) and background consistency (96.48) at 30 seconds are equal to or slightly higher than Self Forcing (97.34, 96.47), not lower.
>
> Regarding the discrepancy with our user study: VBench's consistency metrics are computed from adjacent frame-level feature similarity (DINO for subject, CLIP for background), which is known to penalize appearance changes from camera movement or pose variation even when identity is preserved. Our user study captures perceptual judgment more directly: participants preferred Deep Forcing's subject consistency over all baselines (96.8% vs CausVid, 84.8% vs Self Forcing, 72.2% vs LongLive, 80.0% vs Rolling Forcing).
>
> In the anonymous project page provided in the supplementary.zip file, the videos in the "Comparison with baselines on same prompts" section are all videos used in the user study, and we strongly encourage the reviewer to view the actual videos for comparison.
>
> **W1.** Please refer to our responses to Reviewers PR2k (Q1 & Q2) and zRXT (Q2 & Q3) for detailed analysis on this point, including the unified selection experiment and the three fundamental differences between Deep Forcing and prior sink/compression  approaches.
>
> **W3.** Regarding the extremely high preference rates: the near-100% preference over CausVid reflects a substantial perceptual quality gap. CausVid exhibits noticeable color drift and visual degradation in long-horizon generation, making the preference straightforward for participants, as visible in Figures 6, 18, and 19. Qualitative comparison videos with CausVid are at: https://deeprebuttal.github.io/deeprebutaal/

---

> > ### Author Rebuttal · Reviewer_bodP · 2026-04-03
> >
> > The rebuttal is helpful in several respects. I appreciate the additional analysis suggesting that the increase in dynamic degree is not primarily driven by hallucinated motion, and I also appreciate the clarification that the “LongLive-style” / “Rolling Forcing-style” appendix comparison was intended as a training-free ablation rather than a direct comparison to the fully trained baselines. The explanation for why VBench consistency metrics may diverge from human perceptual judgments is also useful.
> >
> > However, my core concerns remain.
> >
> > First, the novelty concern is still not convincingly resolved. The main mechanisms still appear closely related to known sink / KV-compression ideas from the LLM literature, and this point is not directly established in the present rebuttal. Second, the fairness issue is only partially clarified: even if the appendix comparison was meant as an ablation, we still do not know how strong the training-based baselines would be if retrained with a deeper sink matching the proposed setting. Third, the user-study methodology concern remains. The rebuttal explains why the preference over CausVid can be so strong, but it still does not provide inter-rater agreement or otherwise strengthen the statistical reliability of the study.
> >
> > In my view, these are central issues that would require a more substantial revision and stronger experimental evidence than can be provided in a short rebuttal.

---

> > > ### Author Response · Authors · 2026-04-06
> > >
> > > We appreciate the reviewer's acknowledgment that the previous discussion was helpful. We address the remaining core concerns.
> > >
> > > **First Concern.** We detail the structural differences between our method and each cited LLM technique below (also in our response to Reviewer zRXT(Q3, W1)), demonstrating that Deep Forcing addresses challenges specific to autoregressive video diffusion rather than directly reusing existing methods.
> > >
> > > **(1) Deep Sink(DS) vs. StreamingLLM [1].** StreamingLLM shows that retaining only 4 initial tokens in LLMs is sufficient, and discards all intermediate tokens. Prior video methods (LongLive, Rolling Forcing) borrowed this LLM convention without verifying it for video. We conduct attention analysis on autoregressive video diffusion (Figures 4, 20) and find a different pattern: Self Forcing distributes substantial attention over early and intermediate frames, necessitating deeper sinks (40–60% of the context window).
> > >
> > > **(2) Participative Compression(PC) vs. H2O [2].** H2O retains tokens with the highest cumulative attention scores across the full history. However, in video generation, visual relevance evolves continuously: tokens important earlier may not matter for the current frame. PC instead scores tokens using only the most recent R frames' queries (Eq. 7), directly capturing what the model currently relies on.
> > >
> > > **(3) PC vs. SnapKV [3]***.* SnapKV performs one-shot prompt compression during prefill, fixed by attention patterns within the input prompt before generation begins. Since all tokens generated during decoding are retained without further compression, the cache grows unboundedly with output length. PC operates during decoding, iteratively compressing the rolling KV cache to maintain a bounded window size while adapting to evolving context.
> > >
> > > **(4) PC vs. MorphKV [4].** MorphKV dynamically selects tokens based on already-generated clean tokens. PC additionally leverages the currently denoising query (t=1000) as a selection signal — noisy but meaningful (Table 5; Appendix E), enabling selection that is aware of what the model is about to generate, not just what it has already generated. Furthermore, PC compresses only at the first diffusion timestep when the window overflows, rather than at every chunk generation. After compression, unlike MorphKV, PC applies temporal RoPE unification to the selected tokens to prevent temporal discontinuities.
> > >
> > > In summary, Deep Forcing's novelty lies in: (a) showing that autoregressive video diffusion requires fundamentally deeper sinks, co-designed with importance-aware compression in a 3-partition cache, and (b) addressing video-specific challenges (3D temporal RoPE adjustment, denoising query utilization, diffusion-timestep-aware compression) absent in the LLM setting. We will incorporate the above comparisons and additional clarifications into the paper.
> > >
> > > [1] Xiao et al., Efficient Streaming Language Models with Attention Sinks, ICLR24.
> > >
> > > [2] Zhang et al., H2O: Heavy-Hitter Oracle for Efficient Generative Inference of Large Language Models, NeurIPS23.
> > >
> > > [3] Li et al., SnapKV: LLM Knows What You Are Looking For Before Generation, NeurIPS24.
> > >
> > > [4] Ghadia et al., Dialogue Without Limits: Constant-Sized KV Caches for Extended Responses in LLMs, ICML25.
> > >
> > > **Second Concern.** Our main contribution is a training-free method that matches training-based approaches in video generation performance. Table 1 already evaluates against the fully trained Rolling Forcing and LongLive, where Deep Forcing achieves comparable or better performance. Retraining the Self Forcing with a 10-frame deep sink would require re-running full distillation (trained on 64 H100 GPUs). While we believe this is an interesting research direction, it falls outside the scope of our work — our goal is to achieve competitive performance without any retraining.
> > >
> > > **Third Concern.** We computed Fleiss’ κ for all comparisons and interpret it using the commonly used Landis–Koch convention (1977): 0–0.20 slight, 0.21–0.40 fair, 0.41–0.60 moderate, 0.61–0.80 substantial, 0.81–1.00 almost perfect.
> > >
> > > | vs. Baseline | κ | Agreement Level |
> > > | --- | --- | --- |
> > > | CausVid | ≈1.00 | Almost perfect |
> > > | Self Forcing | 0.83 | Almost perfect |
> > > | LongLive | 0.50 | Moderate |
> > > | Rolling Forcing | 0.38 | Fair |
> > >
> > > The lower κ for LongLive and Rolling Forcing is expected, as these stronger baselines produce finer perceptual differences, naturally leading to more divided individual judgments. Importantly, Deep Forcing was consistently preferred across all four evaluation criteria even in these closer comparisons, showing that the judgments remained directional rather than collapsing to chance-level preferences. We will include these statistics and the Landis-Koch interpretation in the paper.

---

### Official Review · Reviewer_zRXT · 2026-03-09

**Soundness:** 2
**Presentation:** 3
**Significance:** 2
**Originality:** 2
**Overall Recommendation:** 4
**Confidence:** 4

**Summary:**

This paper proposes Deep Forcing, a training-free inference-time framework for improving long-horizon autoregressive video diffusion models. The method introduces Deep Sink, which preserves and temporally realigns early context tokens, and Participative Compression, which selectively prunes KV cache tokens based on attention-based importance. By modifying cache management without additional training, the approach aims to reduce error accumulation and extend generation length to minute-scale videos. Experiments show improvements in motion dynamics and overall consistency while maintaining comparable inference speed.

**Compliance With Llm Reviewing Policy:**

Affirmed.

**Final Justification:**

Overall, I consider this paper to be borderline acceptable, and I am willing to raise my score from 3 to 4.

**Key Questions For Authors:**

1.Deep Forcing is designed to mitigate error accumulation in long-horizon generation, yet in Table 1 (60-second setting) it still underperforms CausVid on subject and background consistency. Given that Deep Forcing is a training-free KV cache strategy built on Self Forcing, can it be directly applied to CausVid? If so, have the authors evaluated this setting?

2.Given that attention sink mechanisms are already widely adopted in recent autoregressive video generation models, can the authors clarify what fundamentally differentiates Deep Sink from prior sink-based approaches beyond scaling and engineering refinements?

3.Why was attention-based importance selected as the compression criterion, and how does it compare to alternative token selection strategies?

4.Can the authors clarify the apparent frame/time labeling inconsistencies in these figures and confirm whether all qualitative comparisons are temporally aligned?

**Limitations:**

yes

**Strengths And Weaknesses:**

Strengths:

1.The proposed method operates entirely at inference time without requiring additional finetuning, making it computationally efficient and easy to integrate into existing autoregressive video diffusion models.

2.The decomposition into DS and PC is conceptually clean, and the mechanism for KV cache management is well-motivated and relatively straightforward to implement.

3.The method demonstrates notable gains in dynamic motion metrics and maintains competitive overall consistency while preserving inference speed, highlighting its practical effectiveness for long-horizon video generation.

Weaknesses:

1.Attention sink mechanisms are already widely adopted in recent autoregressive video generation models, and the proposed method mainly extends this existing paradigm by enlarging the sink and combining it with attention-based KV pruning, making the overall contribution relatively incremental rather than fundamentally novel.

2.As shown in Table 1, the method improves motion metrics but underperforms baselines on subject and background consistency in long-horizon (60-second) settings.

3.Some qualitative figures (e.g., Figure 18 and Figure 19) contain frame or time labeling inconsistencies, which reduce confidence in the clarity and rigor of the evaluation.

4.Participative Compression lacks deeper theoretical justification and comprehensive comparison with alternative compression strategies.

---

> ### Author Rebuttal · Authors · 2026-03-31
>
> **Q1.** We note that VBench's consistency metrics — DINO feature similarity for subject consistency and CLIP feature similarity for background consistency — measure pairwise frame-level feature similarity, which captures local appearance variation between neighboring frames.  As discussed in recent work [1], this gap is particularly pronounced for dynamic videos, where natural appearance changes from camera motion or pose variation increase frame-level feature distance without compromising perceived identity. Due to the characteristics of this metric, CausVid scores high on these consistency metrics, despite producing lower-quality videos, as seen in both the qualitative results and our user studies.  Qualitative comparison videos with CausVid are at: https://deeprebuttal.github.io/deeprebutaal/
>
> We did apply Deep Forcing to CausVid but did not observe consistent improvements, due to a key difference: CausVid performs KV cache recomputation by default, re-forwarding all previous latents through the network at each rollout step. During this process, error-accumulated later latents participate in full attention alongside early latents, propagating errors into the recomputed cache and breaking Deep Sink's core assumption that earliest KV entries remain clean.
>
> [1]: Babu et al. "DynamicEval: Rethinking Evaluation for Dynamic Text-to-Video Synthesis" arXiv25.
>
> **Q2, W1.** Although Deep Sink mainly differs from other methods in the size of the sink, we would like to highlight that it is grounded in a systematic analysis of autoregressive video diffusion that, to our knowledge, has not been conducted before in this domain:
>
> **First systematic attention analysis of Self Forcing.** Prior works (LongLive, Rolling Forcing) adopt 3-frame sinks by borrowing the LLM convention (StreamingLLM) — that only a small number of initial tokens need preservation — without analyzing whether this assumption holds for autoregressive video diffusion. To the best of our knowledge, our work is the first to systematically investigate attention sink mechanisms during inference in autoregressive video generation. Our visualization across diverse layers and heads (Figures 4, 20) reveals a qualitatively different pattern from LLMs: rather than sharp concentration on a few initial tokens, Self Forcing distributes substantial attention over a deep range of early and intermediate tokens. This finding directly motivates a fundamentally larger sink, and is not a design choice that could have been derived from prior LLM-based conventions. We also verified that Deep Forcing applies effectively to the concurrent Causal Forcing[2] model with the same sink ratio, confirming that this finding is not specific to Self Forcing but reflects a broader property of autoregressive video diffusion models. (Causal Forcing results in anonymous Link: https://deeprebuttal.github.io/deeprebutaal/ .)
>
> [2]: Zhu et al. "Causal Forcing: Autoregressive Diffusion Distillation Done Right for High-Quality Real-Time Interactive Video Generation” arxiv26
>
> **Q3, W1.** We selected attention-based importance because the goal is not simply to shrink the KV cache, but to dynamically preserve historical tokens that are still actively relevant to current generation. Our importance score is defined by attention from recent queries to candidate keys, so retained tokens are those the model is actually relying on — making the criterion adaptive and model-aligned, unlike FIFO or fixed sparsity rules.
> We compared against two alternatives: (1) random Top-C selection causes severe artifacts (drift, incoherent transitions — Appendix E, Figure 14), confirming informed selection is necessary; (2) key-similarity selection (cosine similarity between recent and historical keys) consistently underperforms, because feature resemblance alone does not capture functional relevance.
>
> |  | Dyn.Deg | Mot.Sm | OC | IQ | AQ | SC | BC |
> | --- | --- | --- | --- | --- | --- | --- | --- |
> | similarity | 62.15 | *98.10* | *20.47* | *68.78* | *60.25* | *97.04* | *96.31* |
> | Deep Forcing | 57.56 | 98.27 | 20.54 | 69.31 | 60.68 | 97.34 | 96.48 |
>
> We acknowledge that attention-based KV selection exists in the LLM literature, but our contribution is a novel integrated design rather than direct reuse. H2O evicts lowest-scoring tokens globally using accumulated attention score rather than the relevance indicated by the most recent queries. SnapKV performs one-time prompt compression before generation, not for iterative rolling caches. In contrast, Participative Compression is performed by calculating importance through attention scores based on only the most recent queries.
>
> **Q4.**  We thank the reviewer for pointing this out. The videos in Figure 18 are 30-second generations with time labels incorrectly displayed as "60 seconds." We will correct these in the revised paper.
>
> **W4.** Please refer to our response to Reviewer PR2k (Q1&Q2) for a detailed analysis on theoretical justification for our method.

---

> > ### Author Rebuttal · Reviewer_zRXT · 2026-04-03
> >
> > 1. Quantitative metrics: I agree with the authors’ explanation regarding the relatively low VBench consistency scores. In fact, I would suggest using the standard VBench rather than overly smoothed evaluations such as VBench-long. It would also be helpful to include additional metrics, such as color-based measures (e.g., HSV) or frame-level perceptual metrics like LPIPS.
> > 2. Generality: Although DeepForcing does not work on CausVid, it does work on CausalForcing, which alleviates part of my concern.
> > 3. Writing: The authors have committed to correcting the issues related to figures and tables in the experimental section.
> > 4. Method: I still believe that extending Sink frames is somewhat limited in terms of novelty. However, the authors provide a reasonable motivation for this design, and their observations regarding attention mechanisms in video generation models are insightful, which partially addresses my concerns.
> >
> > Overall, I consider this paper to be borderline acceptable, and I am willing to raise my score from 3 to 4.

---

> > > ### Author Response · Authors · 2026-04-07
> > >
> > > We sincerely thank the reviewer for the thoughtful re-evaluation and for raising their score. We are glad that our explanations on attention mechanisms and the motivation behind Deep Sink and Participative Compression addressed the reviewer's concerns. Below, we provide the additional quantitative results as suggested.
> > >
> > > **Standard VBench Results.** We report Standard VBench results, which evaluate single-clip quality without the temporal smoothing of VBench-Long. Unlike in VBench-Long, Deep Forcing outperforms CausVid in both subject consistency and background consistency, which better aligns with the perceptual quality observed in our user study (Table 4). Otherwise, the overall trends are consistent with our VBench-Long evaluations: Deep Forcing achieves the highest dynamic degree while maintaining imaging quality and aesthetic quality comparable to training-based methods (Rolling Forcing, LongLive).
> > >
> > > |  | Dynamic Degree | Motion Smoothness | Overall Consistency | Imaging Quality | Aesthetic Quality | Subject Consistency | Background Consistency |
> > > | --- | --- | --- | --- | --- | --- | --- | --- |
> > > | Rolling Forcing | 32.81 | 98.73 | 21.79 | 70.88 | 62.06 | 93.52 | 94.22 |
> > > | LongLive | 52.34 | 98.74 | 20.93 | 69.46 | 63.41 | 92.09 | 94.03 |
> > > | CausVid | 51.56 | 98.05 | 20.44 | 66.66 | 61.37 | 87.28 | 90.13 |
> > > | Self Forcing | 42.96 | 98.61 | 21.37 | 68.95 | 61.00 | 87.79 | 91.16 |
> > > | Deep Forcing | 60.71 | 98.25 | 21.59 | 69.63 | 62.55 | 91.51 | 93.21 |
> > >
> > > **HSV Color Consistency.** Following the reviewer's suggestion, we additionally measure HSV-based color consistency on 30-second generated videos. We convert each frame to HSV color space and compute a 3D histogram over Hue, Saturation, and Value channels. We report two metrics: (1) **Consecutive Correlation**, the average histogram correlation between adjacent frames, measuring short-term color stability; and (2) **First-Last Correlation**, the histogram correlation between the first and last frames, capturing long-range color drift over the full 30-second duration.
> > >
> > > | Method | Consecutive Correlation ↑ | First-Last Correlation ↑ |
> > > | --- | --- | --- |
> > > | CausVid | 96.68 | 20.64 |
> > > | Self Forcing | 94.12 | 14.93 |
> > > | LongLive | 98.02 | 42.92 |
> > > | Rolling Forcing | 97.25 | 54.27 |
> > > | Deep Forcing | 97.55 | 48.75 |
> > >
> > > Deep Forcing improves HSV color consistency over Self Forcing and CausVid, while remaining competitive with training-based methods such as Rolling Forcing and LongLive. This suggests that our training-free approach effectively preserves color coherence throughout long video generation, mitigating the color drift in Self Forcing.
> > >
> > > Both the Standard VBench and HSV results are consistent with our original analysis, further validating the robustness of our evaluation. We are grateful for the reviewer's constructive suggestions and willingness to re-evaluate our work. We will add these additional results into the paper.

---

### Official Review · Reviewer_PR2k · 2026-03-12

**Soundness:** 3
**Presentation:** 2
**Significance:** 3
**Originality:** 2
**Overall Recommendation:** 3
**Confidence:** 3

**Summary:**

The paper studies long-horizon autoregressive video generation and attributes quality degradation mainly to FIFO KV-cache eviction, which over-emphasizes recent error-corrupted tokens while discarding informative early context. It proposes a training-free cache management strategy built on Self-Forcing. Specifically, Deep Sink keeps a large prefix of early tokens, Participative Compression retains Top-C intermediate tokens based on attention from recent queries, and a temporal RoPE adjustment realigns the retained tokens. Experiments show improved long-horizon consistency and visual quality, with results reported as comparable to training-based baselines on VBench.

**Compliance With Llm Reviewing Policy:**

Affirmed.

**Key Questions For Authors:**

1. Why is it necessary to explicitly keep both sink tokens and recent tokens? Since attention scores typically reflect the importance of KV, could a single sparse selection rule over the whole cache recover the needed sink and recent tokens as well? If not, the paper should explain why.

2. Can the authors give a more principled justification for the overall design, instead of mainly relying on intuition and empirical results?

3. How sensitive is the method to the choice of sink size, recent size, and total cache budget? Do these settings transfer across models and datasets, or do they need to be tuned each time?

4. In the current method, the Top-C KV tokens are selected once and then kept fixed. What would happen if the Top-C selection were updated across different diffusion timesteps? Would this improve the quality of the retained context, or make the method less stable?

**Limitations:**

yes

**Strengths And Weaknesses:**

### Strengths
* The paper is built on a reasonable motivation. It points out that FIFO KV-cache eviction can cause the model to rely too heavily on recently generated, error-prone tokens while losing useful early context. This aligns well with the long-horizon degradation problem.
* The method is simple and training-free, which makes it practically appealing compared with approaches that require additional training. The results are also fairly strong, including competitive comparisons.


### Weaknesses
* The method is quite heuristic. It manually splits the cache into sink tokens, recent tokens, and selected middle tokens, and the temporal RoPE adjustment also feels empirical. Intuition and ablations are not enough analysis for why this specific design is the right one.
* The key hyperparameters are also chosen empirically. In particular, the sink size, recent size, and total budget are tuned, making the method feel more engineered than principled.
* I am not fully convinced that sink tokens and recent tokens need to be preserved explicitly. Since the method already uses attention scores to select important tokens and attention weights can reflect the importance of KV tokens, it is natural to ask whether a single sparse attention selection rule over the KV cache could also recover the needed sink and recent tokens. This point is not clearly addressed.
* The term “KV-cache compression” is somewhat misleading. The method mainly does token selection or pruning, rather than combining or compressing representations in the usual sense.

---

> ### Author Rebuttal · Authors · 2026-03-31
>
> **Q1 & Q2:** Both questions concern our design motivation. Deep Forcing is built on two findings: (1) video diffusion requires a much deeper sink than LLMs, and (2) Deep Sink alone is insufficient for mitigating error during extreme extrapolation, motivating importance-aware pruning.
>
> **1) Deep Sink(DS).** Motivated by StreamingLLM, we analyzed Self Forcing's attention and found that the model attends across a deep range of early and intermediate tokens (Figure 4, Appendix L) — unlike LLMs where only a few initial tokens serve as sinks. This motivates preserving more tokens. This is further justified by the intuition that earlier frames within the training horizon are relatively-error free. Therefore, maintaining a larger early sink helps keep generation closer to the model’s training-time behavior during long-video.
>
> **Can unified selection recover sink tokens?** No. We applied a single attention-score-based selection over the entire cache without explicitly preserving the earliest, least error-accumulated tokens as a fixed sink — instead, we allocated the equivalent budget to additional Top-C selection:
>
> |  | Dyn.Deg | Mot.Sm | OC | IQ | AQ | SC | BC |
> | --- | --- | --- | --- | --- | --- | --- | --- |
> | Unified Selection without DS | 53.00 | 97.58 | 20.35 | 68.79 | 59.51 | 96.41 | 96.06 |
> | Deep Forcing | 57.56 | 98.27 | 20.54 | 69.31 | 60.68 | 97.34 | 96.48 |
>
> We find that this setup performs worse than when using our DS. As Figure 4 shows, attention weights on earliest tokens are not substantially higher than intermediate ones (unlike LLM), so score-based selection ends up evicting the cleaner tokens that would otherwise be cached with our method.
>
> **2) Participative Compression(PC).** While DS effectively mitigates fidelity degradation, it alone cannot fully prevent quality degradation in minute-long generation. This is a well-documented phenomenon in autoregressive long-context generation [1, 2]: beyond training length, the growing KV cache retains increasingly irrelevant tokens, diluting attention across both relevant and irrelevant context and introducing accumulated noise. Recent analysis of video diffusion models shows that attention concentrates on a small subset of semantically critical tokens, with the majority contributing minimally [3] — suggesting that pruning low-attention tokens can reduce error accumulation with limited quality impact. This motivates PC, which dynamically retains only contextually relevant tokens while pruning those that contribute to attention dilution.
>
> **Why preserve recent tokens?** They serve two roles: (a) query proxy for computing Top-C importance scores — without them, PC has no selection basis, and (b) local temporal coherence for smooth frame transitions. Table 5 confirms combining clean past and denoising queries yields optimal selection.
>
> [1]: Ghadia et al. "Dialogue without limits: Constant-sized KV caches for extended responses in LLMs" ICML25.
>
> [2]: Holtzman et al. "The curious case of neural text degeneration" ICLR20
>
> [3]: Yang et al. "Sparse videogen2: Accelerate video generation with sparse attention via semantic-aware permutation” Neurips25
>
> **Q3.** We selected Sink=10, Budget=16, Recent=4 because it yielded the best balanced scores across all VBench dimensions, but neighboring configurations perform comparably. Our method is not brittle to a single tuned configuration; it exhibits a broad stable operating region. We varied each axis independently — Table 6 (Appendix) varies Budget/Recent with Sink=10 fixed, and we additionally varied Sink and Recent with Budget=16 fixed in the link: https://deeprebuttal.github.io/deeprebutaal/ .
>
> Overall, the method works well as long as the sink covers roughly 40–60% of the context window — deeper sinks retain more low-error early tokens, reducing accumulation drift (also supported by Figure 5) — and the recent window is kept close to the current denoising frame, where it most effectively identifies the tokens most relevant to ongoing generation.
>
> **Transferability.** We also applied Deep Forcing to Causal Forcing [4], and observed similar trends in the link: https://deeprebuttal.github.io/deeprebutaal/ .
>
> [4]: Zhu et al. "Causal Forcing: Autoregressive Diffusion Distillation Done Right for High-Quality Real-Time Interactive Video Generation” arxiv26
>
> **Q4:** We performed Top-C selection independently at every diffusion timestep:
>
> |  | Dyn.Deg | Mot.Sm | OC | IQ | AQ | SC | BC |
> | --- | --- | --- | --- | --- | --- | --- | --- |
> | Per-Timestep | 59.16 | 98.18 | 20.52 | 69.35 | 60.01 | 97.28 | 96.51 |
> | Deep Forcing | 57.56 | 98.27 | 20.54 | 69.31 | 60.68 | 97.34 | 96.48 |
>
> Nearly identical performance, as expected from Figure 12 (Appendix D) — attention patterns across timesteps are highly consistent, so tokens important at the initial timestep remain important throughout denoising. Meanwhile, per-timestep selection requires maintaining separate KV caches for each timestep, leading to additional VRAM consumption.

---

> > ### Author Rebuttal · Reviewer_PR2k · 2026-04-03
> >
> > The rebuttal provides a clearer motivation for explicitly preserving sink and recent tokens, and the additional ablations are helpful. However, my main concern remains that the overall design is still largely heuristic, with key components and hyperparameters justified primarily by intuition and empirical tuning rather than by a more principled analysis.

---

> > > ### Author Response · Authors · 2026-04-06
> > >
> > > We thank the reviewer for acknowledging our added motivation and ablations. We address the remaining concern below.
> > >
> > > We would like to separate two aspects of our method: the **overall design** ([sink | compressed | recent] partition) and the **hyperparameters** (specific numerical values), as we believe they warrant distinct discussion.
> > >
> > > **[sink | compressed | recent] partition is motivated by our attention mechanism analysis specific to autoregressive video diffusion models.**
> > >
> > > - **Sink tokens** are retained because the model distributes substantial attention across a deep range of early and intermediate tokens (Figures 4 and 20), unlike LLMs where only 1–4 initial tokens act as sinks. This deep attention pattern, consistent across layers and heads, motivates preserving a larger early context than what LLM-based methods require.
> > > - **Recent tokens** are retained for two functional reasons: (a) they provide the query basis for computing Top-C importance scores — without them, importance-based selection has no selection signal, and (b) they ensure local temporal coherence for smooth frame transitions. Figures 4 and 20 consistently show that the most recent 3–4 frames receive high attention across layers and heads.
> > > - **Compressed middle** tokens occupy the region between the early context and the recent frames. Our analysis shows that tokens selected by attention-based importance consistently align with semantically meaningful regions such as object bodies, articulated parts, and persistent scene structure (Fig. 13), while replacing this selection with random Gaussian selection causes immediate context drift and severe artifacts (Fig. 14, 15) — confirming that the selection criterion carries genuine signal. Even a reasonable alternative — key-similarity-based selection using cosine similarity — consistently underperforms, because feature resemblance alone does not capture functional relevance to ongoing generation. These observations are consistent with findings on attention dilution in long-context generation [1, 2], attention sparsity in video diffusion [3], and a similarly-motivated attention-based selection principle in LLM KV cache methods [5, 6, 7].
> > >
> > > **Hyperparameter values are empirically determined, but grounded in attention mechanism analysis.**
> > >
> > > We acknowledge that the specific numerical values (Sink=10, Budget=16, Recent=4) are empirically selected. However, this is the standard practice across KV cache methods at top venues:
> > >
> > > - StreamingLLM [4] (ICLR'24): sink size = 4, selected empirically as sufficient without theoretical derivation.
> > > - H2O [5] (NeurIPS'23): heavy-hitter budget = 20%, selected by sweeping 4%–60% cache ratios. The budget ratio itself is empirically chosen.
> > > - SnapKV [6] (NeurIPS'24): observation window size (16 or 32), pooling kernel size (5 or 7), and cache budget described as "hyperparameters that can be customized."
> > > - MorphKV [1] (ICML'25): cache size, recent window size empirically selected, with the authors noting "MorphKV shows minimal sensitivity to variations in its hyperparameters."
> > >
> > > None of these methods derives its specific values from first principles. They follow the similar pattern: attention analysis, a structurally motivated design, validated by empirical experiments within a reasonable range. Deep Forcing follows the same practice: structurally motivated design validated by empirical ablation. We further demonstrate robustness to hyperparameter choices  and generalization to other models (Causal Forcing, Matrix-Game 2.0).
> > >
> > > To summarize, while our design and hyperparameters are empirically determined, they are motivated by well-documented phenomena in autoregressive long-context generation [1, 2] and attention sparsity in video diffusion [3], and grounded in attention analysis (Figures 4, 12, 20). Their effectiveness is validated through comprehensive ablations and analyses (Tables 5, 6; Appendix A, D, E). Therefore, we believe this constitutes sufficient evidence that our design is both well-grounded and effective.
> > >
> > > [4] Xiao et al., Efficient Streaming Language Models with Attention Sinks, ICLR24.
> > >
> > > [5] Zhang et al., H2O: Heavy-Hitter Oracle for Efficient Generative Inference of Large Language Models, NeurIPS23.
> > >
> > > [6] Li et al., SnapKV: LLM Knows What You Are Looking For Before Generation, NeurIPS24.

---

### Decision · Program_Chairs · 2026-04-30

**Decision:**

Accept (regular)

**Comment:**

This paper proposes to improve autoregressive long-video generation by better managing context tokens at inference time. Early tokens are preserved and re-aligned. KV cache tokens are selectively pruned. Empirical results demonstrate that the proposed method can extend high-quality generation length.

After the rebuttal and discussion, the reviewers' ratings are mixed, with two reviewers recommending acceptance and two recommending rejection.

The rebuttal resolved Reviewer zRXT's questions on the evaluation and generality, as well as other clarification questions from Reviewer HoYe.

Reviewer bodP kept a weak rejection rating after discussion, citing the limited technical contribution beyond attention sinks and KV-cache selection/pruning, and the lack of comparison to training-based methods. Reviewer PR2k recommended a weak rejection, citing the heuristic categorization of tokens and the lack of principled analysis of some hyperparameters.

The AC agreed with the reviewers that the proposed method consists of quite a few heuristic decisions and can be improved by incorporating comparison to some training-based methods. However, given its current state, the proposed cache management techniques are well-motivated and achieve good empirical results. Sharing them with the community could be beneficial.

The decision is to recommend the paper for acceptance. The authors are encouraged to revise the paper to incorporate the comments from the rebuttal period in the final version.